# Position: Don't Use the CLT in LLM Evals With Fewer Than a Few Hundred Datapoints

**Sam Bowyer** [1]  **Laurence Aitchison** [*1]  **Desi R. Ivanova** [*2]

## Abstract

Rigorous statistical evaluations of large language models (LLMs), including valid error bars and significance testing, are essential for meaningful and reliable performance assessment. Currently, when such statistical measures are reported, they typically rely on the Central Limit Theorem (CLT). In this position paper, we argue that while CLT-based methods for uncertainty quantification are appropriate when benchmarks consist of thousands of examples, they fail to provide adequate uncertainty estimates for LLM evaluations that rely on smaller, highly specialized benchmarks. In these small-data settings, we demonstrate that CLT-based methods perform very poorly, usually dramatically underestimating uncertainty (i.e. producing error bars that are too small). We give recommendations for alternative frequentist and Bayesian methods that are both easy to implement and more appropriate in these increasingly common scenarios. We provide a simple Python library for these Bayesian methods at https://github.com/sambowyer/bayes_evals.

## 1. Introduction

Benchmarks provide a systematic way for measuring the capabilities and risks of large language models (LLMs), for tracking progress over time as well as enabling performance comparison across different models. Such language model evaluations ("LLM evals") inform critical decisions about model selection and deployment. However, current benchmarking practices rarely quantify the inherent statistical uncertainty in these evals, which can substantially undermine

the validity of and confidence in the reported results (Marie et al., 2021; Reuel et al., 2024; Biderman et al., 2024).

Recent works have recognized the importance of statistical rigour in LLM evals and the need to improve it, for instance, through the inclusion of error bars (Dror et al., 2018; Miller, 2024; Madaan et al., 2024; Hermann et al., 2024). When such uncertainty estimates are reported at all, they are most often asymptotic, based on Central Limit Theorem (CLT).

In this position paper, we take it as read that LLM evals should come with error bars. Instead, we ask about the best way to compute those error bars. **Here, we argue that while CLT-based methods work well in LLM evals with thousands of examples, they systematically fail to provide valid uncertainties for smaller, specialized benchmarks, which are becoming increasingly common. In these settings, accurate uncertainty quantification requires more appropriate frequentist or Bayesian methods.**

Many prominent LLM benchmarks such as Big Bench (Srivastava et al., 2023), MMLU (Hendrycks et al., 2021a), GSM8K (Cobbe et al., 2021), have large evaluation sets, on the order of hundreds to thousands. Such benchmarks focus on relatively straightforward tasks that many LLMs have largely saturated (e.g. high school science questions), and generally do not accurately represent the tasks LLMs are used for in practical real-world applications (Raji et al., 2021; Hardy et al., 2025). In contrast, both industry practitioners developing proprietary benchmarks, and researchers evaluating advanced capabilities of frontier models, such as advanced reasoning, multi-turn tool use or tasks involving specialized domain expertise, increasingly focus on more targeted and representative benchmarks with very high quality labels. These benchmarks are much more expensive to construct and therefore tend to involve far fewer examples, often on the order of tens to hundreds per task.

There are plenty of examples that illustrate this trend. For instance, CUAD (Hendrycks et al., 2021b) is a specialized contract understanding dataset that consists of 510 labelled documents across 25 contract types. It required extensive annotation efforts from law students and reviews from experienced attorneys, with an estimated cost of around $2 million. FrontierMath (Glazer et al., 2024; Pillay, 2024)—

*Equal contribution  [1]University of Bristol, UK  [2]University of Oxford, UK. Correspondence to: Sam Bowyer <sam.bowyer@bristol.ac.uk>, Laurence Aitchison <laurence.aitchison@gmail.com>, Desi R. Ivanova <desi.ivanova@stats.ox.ac.uk>.

*Proceedings of the $42^{nd}$ International Conference on Machine Learning*, Vancouver, Canada. PMLR 267, 2025. Copyright 2025 by the author(s).

a new benchmark developed through a collaboration with over 60 expert mathematicians—contains around 300 problems across 23 categories, some of which have fewer than 3 samples. Other benchmarks relevant to today's frontier models include AIME with 15 competition math problems, SWE Bench Verified, containing 500 samples across 4 difficulty levels (Jimenez et al., 2024), MLE Bench with 75 Kaggle competitions (Chan et al., 2025), and LiveBench, which frequently updates tasks across 6 categories, currently averaging 55 examples per task (White et al., 2025).

Furthermore, even large benchmarks such as Big Bench are often broken down into smaller sub-tasks. As data from two different sub-tasks cannot possibly be considered independent and identically distributed (IID), we cannot naively use the size of the overall dataset to justify applying the CLT. Instead, each sub-task should be treated separately, and because these sub-tasks are typically much smaller, CLT-based approaches again become unreliable.

In this paper, we show that CLT-based confidence intervals can be extremely problematic in these small data regimes. To make robust comparisons against alternative methods, both frequentist and Bayesian, we conduct a large suite of experiments with realistic simulated data, where the true parameter values are known. We construct various intervals and measure their *coverage*, that is, the proportion of times those intervals contain the ground truth. We find that, particularly with smaller datasets, the CLT produces unreliable intervals that fail to achieve their target confidence level, also known as *nominal coverage*. This is true across a variety of settings: when evaluating a single model and when comparing two models, both with IID and clustered questions.

In simpler settings, such as computing error bars for a single model on IID data, we find that there are more appropriate, non-CLT-based frequentist methods that perform as well as Bayesian methods. However, in more complex settings, such frequentist methods are less readily available, while Bayesian methods allow us to easily extend to cases such as clustered data (e.g. the reading comprehension task discussed above) or when dealing with arbitrary metrics that are not averages of IID variables (e.g. $F$-scores).

## 2. The Central Limit Theorem and Frequentist Uncertainty Quantification

The Central Limit Theorem is a foundational result in frequentist statistical inference. It states that, for sufficiently large sample size $N$, the sampling distribution of the sample mean will be approximately normal, regardless of the distribution of the original population.

Formally, if $X_1, \ldots, X_N$ are **independent and identically distributed** random variables with mean $\mu \in \mathbb{R}$ and finite

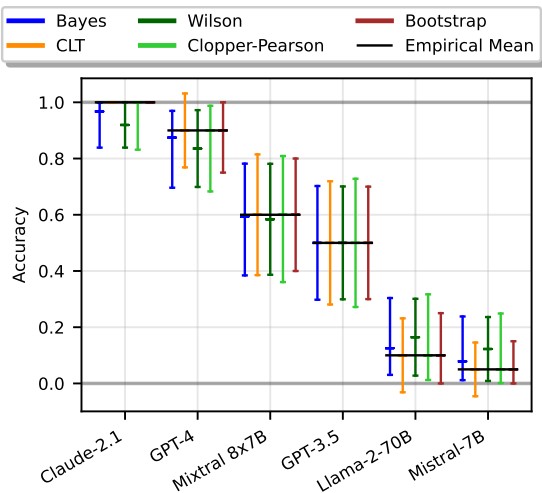

Figure 1: **Error bars on LangChain tool-use benchmark.** 95% intervals for model accuracy on $N=20$ questions. The CLT produces invalid intervals, extending beyond $[0, 1]$ or collapsing to zero, highlighting its unreliability in practical settings. The alternative frequentist and Bayesian methods we advocate yield valid and well-calibrated intervals even in this small-data regime. See Appendix B for more results.

variance $\sigma^2 > 0$, then

$$\sqrt{N}(\hat{\mu} - \mu) \xrightarrow{d} \mathcal{N}\left(0, \sigma^2\right) \text{ as } N \to \infty, \quad (1)$$

where $\hat{\mu} = \frac{1}{N} \sum_{i=1}^{N} X_i$ is the sample mean. Put differently, the distribution of $\hat{\mu}$ is arbitrarily close to $\mathcal{N}(\mu, \frac{\sigma^2}{N})$ for large enough $N$. The result extends verbatim to the multivariate setting: if each $X_i$ is a $p$-dimensional random vector with mean $\mu$ and covariance $\Sigma$, then $\hat{\mu} = \frac{1}{N} \sum_{i=1}^{N} X_i$ is approximately $\mathcal{N}(\mu, \frac{\Sigma}{N})$ then for large $N$ (Van der Vaart, 2000).

The CLT underlies many common frequentist methods for uncertainty quantification, such as confidence intervals and hypothesis tests. To apply it in practice, however, we need to know population variance $\sigma^2$. Of course we rarely know $\sigma^2$ and so we estimate it empirically by the sample variance, $S^2 = \frac{1}{N-1} \sum_{i=1}^{N} (X_i - \hat{\mu})^2$. By Slutsky's theorem, replacing $\sigma^2$ with its *consistent* estimator $S^2$ in Eq. 1 preserves the asymptotic normality (see e.g. Theorem 11.3.2 in Lehmann et al., 2022), so that we have $\sqrt{N}(\hat{\mu} - \mu) \approx \mathcal{N}(0, S^2)$.

The standard error of the sample mean, $\mathrm{SE}(\hat{\mu})$, is the standard deviation of the sampling distribution of $\hat{\mu}$, and equals $\mathrm{SE}(\hat{\mu}) = \sqrt{S^2/N}$.

### 2.1. CLT-Based Confidence Intervals

For a desired confidence level $1 - \alpha$, typically 95% ($\alpha = 0.05$) or 99% ($\alpha = 0.01$), the general form of a two-sided CLT-based confidence interval (CI) for the mean $\mu$ is

$$\mathrm{CI}_{1-\alpha}(\mu) = \hat{\mu} \pm z_{\alpha/2}\, \mathrm{SE}(\hat{\mu}), \quad (2)$$

where $z_{\alpha/2}$ is the $\alpha/2$-th quantile of the standard normal distribution (e.g. $z_{0.025} \approx 1.96$ for a 95% CI).

**Two independent samples** Often we want to compare the means $\mu_A$ and $\mu_B$ from two independent samples with sizes $N_A$ and $N_B$ and sample variances $S_A$ and $S_B$. The parameter of interest is then usually their difference $\mu_A - \mu_B$. Under the independence assumption, the variance of the difference in sample means is the sum of their individual variances, so the standard error of the difference is $\mathrm{SE}(\hat\mu_A - \hat\mu_B) = \sqrt{S_A^2/N_A + S_B^2/N_B}$. The $1 - \alpha$ confidence interval for $\mu_A - \mu_B$ then follows the same template as Eq. 2:

$$\mathrm{CI}_{1-\alpha}(\mu_A - \mu_B) = (\hat\mu_A - \hat\mu_B) \pm z_{\alpha/2} \, \mathrm{SE}(\hat\mu_A - \hat\mu_B). \quad (3)$$

**Two paired samples** In many scenarios, the two samples are *paired*, meaning each observation in sample $A$ is naturally matched with one in sample $B$. Let $(X_{A,i}, X_{B,i})$ for $i = 1, \ldots, N$ and define the difference $D_i = X_{A,i} - X_{B,i}$. The parameter of interest for which we wish to construct a confidence interval is $\mu_D = \mathbb{E}[D_i]$. This setup arises often, for example in *before-after* studies where measurements are taken on the same subjects before and after a treatment or models are evaluated before and after interventions such as RLHF. The CLT applies directly to the differences $D_1, \ldots, D_N$, which we treat as a single sample and the form of the confidence interval the same as in Eq. 2:

$$\mathrm{CI}_{1-\alpha}(\mu_D) = \hat\mu_D \pm z_{\alpha/2} \, \mathrm{SE}(\hat\mu_D). \quad (4)$$

### 2.2. CLT-Based Hypothesis Testing

There is a direct correspondence between null hypothesis significance testing (NHST) and confidence intervals: for a two-sided test at significance level $\alpha$, we reject $H_0$ if and only if the $1 - \alpha$ confidence interval not contain zero. The relationship extends to one-sided tests with appropriate modifications (see §3.5, 5.4 in Lehmann et al., 2022).

To illustrate this connection, consider testing whether a population mean $\mu$ equals some hypothesized value $\mu_0$. The classical setup defines a null hypothesis $H_0 : \mu = \mu_0$ and an alternative, $H_1 : \mu \neq \mu_0$ (or a one-sided e.g. $H_1 : \mu > \mu_0$). By the CLT and Slutsky's Theorem, we have that

$$T_{\text{one-sample}} = \frac{\hat\mu - \mu_0}{\mathrm{SE}(\hat\mu)} = \frac{\sqrt{N}(\hat\mu - \mu_0)}{S}$$

follows a standard normal distribution asymptotically. We *reject* the null hypothesis $H_0$ at a pre-specified significance level $\alpha$ (usually $\alpha = 0.05$ or $0.01$) if $|T| > z_{\alpha/2}$ for the two-sided test, or if $T > z_\alpha$ (or $T < -z_\alpha$) for the one-sided test. Otherwise, we *fail to reject* $H_0$. This is exactly equivalent to checking if the $(1 - \alpha)$ confidence interval excludes $\mu_0$.

Similarly, when comparing two sample means (independent or paired), testing whether their difference ($\mu_A - \mu_B$ or $\mu_D$) is zero is equivalent to checking if zero lies within the CI.

This equivalence means that both approaches share the same assumptions and limitations, particularly their reliance on asymptotic normality and independent sampling, making them equally unreliable in the small data regimes.

**Special case: Bernoulli data** When $X_i \overset{\text{IID}}{\sim} \text{Bernoulli}(\theta)$, the variance of the distribution is completely determined by the mean: $\mathrm{Var}(X_i) = \theta(1 - \theta)$. Using its sample estimate $S^2 = \hat\theta(1 - \hat\theta)$, the standard error simplifies to

$$\mathrm{SE}(\hat\theta) = \sqrt{\frac{\hat\theta(1 - \hat\theta)}{N}}, \quad \text{where } \hat\theta = \frac{1}{N}\sum_{i=1}^{N} X_i. \quad (5)$$

This gives us the well-known formula for the confidence interval of a proportion, which has been recommended by and used in several recent works for LLM evals (e.g. Madaan et al., 2024; Miller, 2024; Dubey et al., 2024).

## 3. Failures of the CLT for LLM Evals

We show that CLT-based CIs break down in many settings in the context of LLM evals with small sample sizes. Each subsection includes an experiment presenting a different failure mode, following the evaluation protocol outlined below.

**Experimental setup** In each experiment, we sample 100 values of the underlying model performance parameter, $\theta$, from a specified prior distribution. For each $\theta$, we generate 200 independent datasets of sizes $N = 3, 10, 30$ and 100, giving us a total of 80,000 LLM eval datasets. This large number of datasets ensures that our results are robust to the randomness in the data generation procedure. We additionally repeated experiments five times with different random seeds (affecting both the sampled parameters and datasets), resulting in standard errors in coverage on the order of $10^{-3}$ or lower (see Appendix G). For every dataset, we construct a range of *interval methods* for 100 different nominal coverage levels, $1 - \alpha$, ranging from 0.8 to 0.995.

**Interval methods** In all experiments, we use at least three methods to construct intervals:

- **CLT-based** confidence intervals, as described in § 2.1.
- **Bootstrap** confidence intervals which are obtained by resampling the original data $K$ times (with replacement) to generate a distribution of the estimator $\{\hat\theta^{(k)}\}_{k=1}^{K}$ and taking the empirical $\alpha/2$ and $1 - \alpha/2$ quantiles to form the interval. Since bootstrap performance depends greatly on $K$ (Davidson & MacKinnon, 2000), we use a large $K = 10,000$ throughout.
- **Bayesian** credible intervals derived from the posterior of $\theta$, computed either exactly for conjugate models or approximated using importance sampling. Credible intervals are not unique; in our experiments, we report quantile-based intervals (QBI), either analytic or empirical. Highest posterior density intervals (HDI) is a common alternative which we ablate in Appendix J.

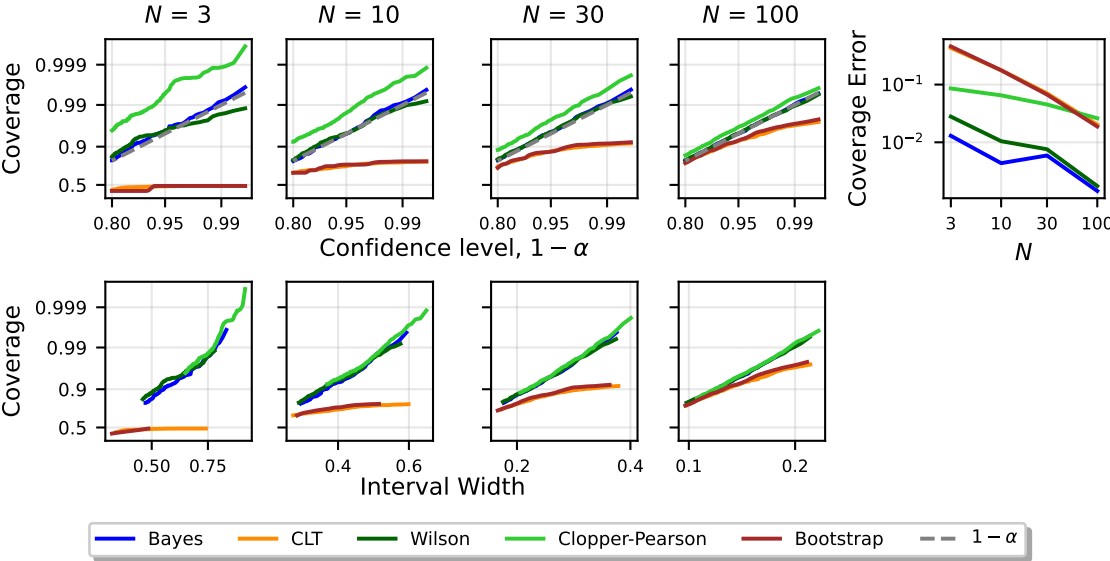

Figure 2: **IID question setting.** Coverage vs. confidence level (**top**) and vs. interval width (**bottom**) for various interval-calculation methods on the value of $\theta$. While all methods approach the ideal $1 - \alpha$ coverage line for large $N$, only the Bayesian credible interval and Wilson confidence intervals achieve this for small $N$.

**Evaluation metrics** Frequentist confidence intervals and Bayesian credible intervals are fundamentally different in their interpretation. A 95% confidence interval means that under repeated sampling (as we do here), 95% of the computed intervals will contain the true $\theta$. A 95% credible interval has a 95% (posterior) probability of containing $\theta$. Indeed, confidence intervals are commonly misinterpreted as credible intervals (Hubbard, 2011; Greenland et al., 2016).

Despite these differences in interpretation, we evaluate both methods on the same frequentist criterion—*coverage probability*, which we estimate as the empirical proportion of intervals that contain the true $\theta$. To highlight differences at the high confidence levels (i.e., 95% and above), which are most relevant in practice, we use logit-logit axes when plotting empirical versus nominal coverage. We also report mean absolute distance from nominal coverage, averaged across $\alpha$ and experiment repeats, which we refer to as 'coverage error'; this is shown on log-log axes against sample size $N$. We additionally record the average interval width, which we report in Appendix H, along with ablations on the choice of the data generating prior in Appendix I.

### 3.1. Failure of CLT-Based Confidence Intervals in IID Questions Setting

It is common to assume that benchmarks contain IID questions, so that each eval outcome is a Bernoulli trial with some underlying probability of success $\theta$. Under this assumption, we compute the standard standard error of the empirical mean $\hat{\theta}$ using Eq. 5. However, this expression raises an immediate problem: as $\hat{\theta}$ approaches 0 or 1, the confidence interval shrinks towards 0, incorrectly suggesting

certainty. In small-data settings, it is perfectly possible that the model gets all questions right ($\hat{\theta} = 1$) or wrong ($\hat{\theta} = 0$). In either case, the CLT-based interval (Eq. 2) vanishes, since $\text{SE}(\hat{\theta}) = 0$, which is clearly not right. Additionally, when $\hat{\theta}$ or $1 - \hat{\theta}$ is less than $1/(1 + N/z_\alpha^2)$, the boundaries of the interval would fall outside the valid $[0, 1]$ range. As Fig. 1 shows, both of these problems do occur in practice.

While these issues might occur in practice only rarely, they do highlight that the assumptions underlying the asymptotic, CLT-based approaches may not be suitable for LLM evals, at least in smaller data regimes. In that case, we expect to see broader failures of CLT-based confidence intervals.

To empirically test this hypothesis, we evaluate the coverage of different types of intervals by generating data from

$$\theta \sim \text{Beta}(1, 1) \quad y_i \sim \text{Bernoulli}(\theta) \text{ for } i = 1, \ldots N. \quad (6)$$

This simulation framework mimics the common LLM eval scenario described at the start of this subsection, with true model accuracy $\theta$ uniformly distributed between 0 and 1.

In addition to the three primary interval methods, here we consider two additional frequentist approaches designed specifically for Bernoulli trials: the approximate Wilson score interval (WS, Wilson, 1927) and the exact Clopper-Pearson interval (CP, Clopper & Pearson, 1934), which we describe in Appendix A for completeness.

The results are shown in Fig. 2: the top row plots the confidence level against the empirical coverage. We find that both CLT-based and bootstrap CIs show poor calibration, as the actual coverage is well below the nominal $1 - \alpha$ level, indicated by the gray dashed line. This is a fairly catastrophic failure: the whole point of a confidence interval, by

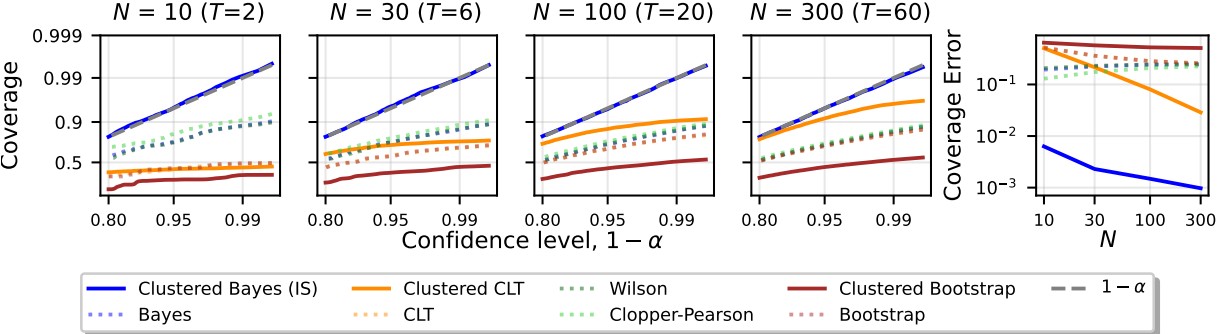

Figure 3: **Clustered questions setting**. Coverage vs. confidence level for various interval-calculation methods on the value of $\theta$. See Appendix H for interval widths. Importantly, note that in a small-data regime, neither simple CLT nor clustered CLT intervals produce correct coverage. Methods ignoring the clustered structure of the data are shown as dotted lines.

construction, is to get the right coverage. As an example, in the $N=100$ column we see that 95% CLT-based intervals achieve a coverage of only 92.5%. This leads to a significant difference in interval width: the corresponding Gaussian quantiles (used in Eq. 2) for confidence 0.95 and 0.925 are $z_{0.025} \approx 1.96$ and $z_{0.0125} \approx 1.78$. Furthermore, looking at the width of these intervals (bottom row in Fig. 2), we find that CLT-based or bootstrap CIs are inefficient—they are wider than needed for any given coverage.

In contrast, we find that simple Bayesian methods based on a Beta-Bernoulli distribution, and the frequentist WS interval perform well. While approximate, WS demonstrates favourable coverage and length properties and is generally preferred to the exact CP, which is overly conservative (too wide) in practice (Agresti & Coull, 1998; Newcombe & Nurminen, 2011). Interestingly, the CP interval can be shown to be equivalent to the Bayesian credible interval but with the uniform prior removed (Thulin, 2014). In other words, the Bayesian approach can be seen as providing a form of shrinkage of the CP interval thus mitigating over-coverage.

We would, therefore, recommend using WS or Bayesian intervals in practice. Both methods are easy to apply. WS (and CP) are implemented in SciPy. The posterior for Beta-Bernoulli is available in closed form:

$$\mathbb{P}(\theta|y_{1:N}) = \text{Beta}\left(1 + \sum_{i=1}^{N} y_i, 1 + \sum_{i=1}^{N}(1 - y_i)\right), (7)$$

so we can use the quantiles of the Beta distribution:

```
Snippet 1: Analysis for a single model
1  from scipy.stats import binomtest, beta
2
3  # y is a length N binary "eval" vector
4  S, N = y.sum(), len(y) # total successes & questions
5  result = binomtest(k=S, n=N)
6
7  # 95% Wilson score and Clopper-Pearson intervals
8  wilson_ci = result.proportion_ci("wilson", 0.95)
9  cp_ci = result.proportion_ci("exact", 0.95)
10
11 # Bayesian Credible interval
12 posterior = beta(1+S, 1+(N-S))
13 bayes_ci = posterior.interval(confidence=0.95)
```

## 3.2. Failure of CLT-Based Confidence Intervals in Clustered Questions Setting

The previous discussion was based on the simplest setting: one LLM and IID questions. Miller (2024) emphasizes that the CLT-based approach is flexible enough to apply to more complex settings, such as when questions are clustered. Examples include reading comprehension benchmarks that have multiple questions about a single passage of text (e.g. Dua et al., 2019; Choi et al., 2018; Lai et al., 2017; Rajpurkar et al., 2018; Shi et al., 2023). We would expect that LLMs might find some passages of text easier to understand than others, which would result in varying performance—better accuracy on simpler passages and lower accuracy on more complex passages. Importantly, this introduces non-IID structure in the responses which must be accounted for. To address this, Miller (2024, Sec. 2.2) suggests using clustered standard errors (Abadie et al., 2023). This is a post-hoc adjustment to account for the correlation among the $T$ clusters with $N_t$ questions each and $\sum_{t=1}^{T} N_t = N$:

$$\text{SE}_{\text{clust.}} = \sqrt{\text{SE}_{\text{CLT}}^2 + \frac{1}{N^2}\sum_{t=1}^{T}\sum_{i=1}^{N_t}\sum_{j\neq i}^{N_t}(y_{i,t} - \bar{y})(y_{j,t} - \bar{y})}.$$

Here $y_{i,t} \in \{0, 1\}$ is the success on question $i$ of task $t$, and $\bar{y} = \frac{1}{N}\sum_{t=1}^{T}\sum_{i=1}^{N_t} y_{i,t}$. To assess the effectiveness of this approach, we use data from the following generative model:

$$d \sim \text{Gamma}(1, 1), \quad \theta \sim \text{Beta}(1, 1)$$
$$\theta_t \sim \text{Beta}(d\theta, d(1 - \theta)), \quad y_{i,t} \sim \text{Bernoulli}(\theta_t). \quad (8)$$

Here, $d$ controls the the range of difficulties of the tasks or clusters (i.e. the *concentration*), $\theta$ is the global performance of the model, and is the quantity we are trying to infer, while $\theta_t$ is the performance on a given task. Note that $\mathbb{E}[\theta_t | \theta] = \theta$, so $\theta$ controls the expected accuracy on any given task. If $d$ is large then $\theta_t$ is always close to $\theta$, which indicates lower correlations between questions within a task. In contrast, if $d$ is small, then $\theta_t$ is further from $\theta$, implying larger correlations between questions in a task/cluster.

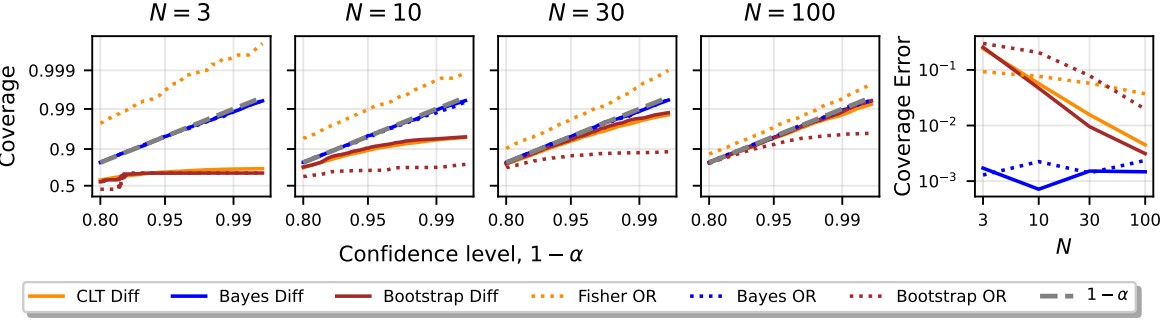

Figure 4: **Independent model comparison setting**. Coverage vs confidence level for various interval-calculation methods when comparing two independent means $\theta_A$ and $\theta_B$ for both the difference (Diff) and odds ratio (OR) metrics. The diagonal gray dashed line represents the expected coverage, $1 - \alpha$. The CLT is not applicable to the OR.

To perform inference in this model, we can integrate out $\theta_t$ so that the total number of correct answers in task $t$ is given by $Y_t \sim \text{BetaBin}(N_t, d\theta, d(1 - \theta))$. For simplicity, we use importance sampling (IS) with the prior as proposal: $\theta \sim \beta(1, 1)$ and $d \sim \text{Gamma}(1, 1)$. Then the importance weights $\{w^{(k)}\}_{k=1}^K$ are given by the Beta-Binomial likelihood of the data $Y_t$ under each $(\theta^{(k)}, d^{(k)})$:

$$w^{(k)} = \prod_{t=1}^T \text{BetaBin}(Y_t; N_t, d^{(k)}\theta^{(k)}, d^{(k)}(1 - \theta^{(k)})).$$

We use these weights to resample (with replacement) our collection of samples $\{\theta^{(k)}\}_{k=1}^K$ and calculate credible intervals using the relevant percentiles in the resulting collection. Simple Python code for this IS is provided in Appendix E.1.

Fig. 3 compares the three primary methods under both the IID and clustered question assumption along with the CP and WS intervals (which also assume IID data). To our knowledge there are no readily available frequentist methods, tailored to such clustered data that can be applied in here. Of all interval calculation methods we consider, only the Bayesian method based on a clustered model (Eq. 8) achieves the right coverage across different sample sizes.

### 3.3. Failure of CLT-Based Confidence Intervals in Independent Model Comparison Setting

We often wish not only to construct a confidence interval for the performance of a single model, but also to compare two models. Consider the setup from § 3.1 but now we have two language models, $A$ and $B$. Let $\theta_A$ and $\theta_B$ be the true probabilities of success of models $A$ and $B$, each of which is independently generating Bernoulli outcomes as in the model defined in Eq.6. This setup is applicable, for example, when we only have access to the empirical accuracies, $\hat{\theta}_A$ and $\hat{\theta}_B$, and not the per-question binary data, itself ($y_{A;i}$ and $y_{B;i}$), or when models are evaluated on different sets of questions. In these cases, there are two main approaches for comparing $\theta_A$ and $\theta_B$:

- looking at their *difference*, $\text{Diff} = \theta_A - \theta_B$, and checking if 0 lies within its $(1 - \alpha)$ confidence interval.
- looking at their *odds ratio*, $\text{OR} = \frac{\theta_A/(1-\theta_A)}{\theta_B/(1-\theta_B)}$, and checking if 1 lies within its $(1 - \alpha)$ confidence interval.

A confidence interval including the respective null value (0 for the difference, 1 for the odds ratio) suggests that $\theta_A$ and $\theta_B$ are statistically indistinguishable at level $\alpha$.

With the CLT, we can only construct a confidence interval on the difference in performances, since the odds ratio is a non-linear transformation of the parameters. Whilst the CLT guarantees asymptotic normality for the difference of proportions, it does not extend to their ratio or odds ratio (more on this in § 3.5). As shown by the solid orange curves in Fig. 4, the CLT-based interval from Eq. 3 has coverage far below the target level when $N$ is small. Although better frequentist methods exist, e.g. the hybrid score interval introduced in Newcombe (1998), they are harder to implement and are not available in standard libraries.

For odds ratio analysis, the standard frequentist method to obtaining confidence intervals involves inverting the Fisher's exact test (FET, Fisher, 1922), which *guarantees* coverage of at least $1 - \alpha$ at any dataset size and any pair of parameters. However, similarly to the CP exact interval for a single proportion, FET tends to be overly conservative, especially for small $N$, as shown by the dotted orange curves in Fig.4.

The Bayesian approach is able to give us credible intervals for both the difference and the odds ratio. To construct these, we draw samples from the exact posterior (Eq.7), compute the metric of interest and take the empirical quantiles. As Fig.4 shows, the Bayesian intervals achieves excellent coverage across all sample sizes for both metrics.

Altham (1969) showed that Fisher's exact test corresponds to our Bayesian analysis if we use highly conservative priors, $\theta_A \sim \text{Beta}(1, 0)$ and $\theta_B \sim \text{Beta}(0, 1)$. These improper priors effectively assume the most extreme scenario—perfect

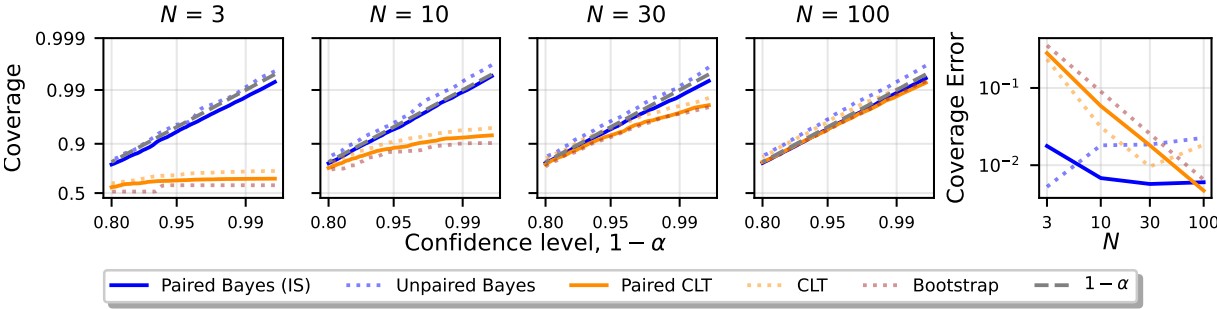

Figure 5: **Paired model comparison setting**. Coverage vs. confidence level for various interval-calculation methods on the value of $\theta_A - \theta_B$. Methods ignoring the paired structure of the data—assuming instead IID questions and answers from model A and from model B, as per § 3.1—are shown as dotted lines.

performance for model A and worst possible performance for model B, which helps explain the over-coverage.

With small data, we recommend Bayesian intervals for the difference or odds ratio. It can be implemented as follows:

```
Snippet 2: Bayesian analysis: Model comparison

1  # y_A and y_B are vectors of evals for two models
2  import numpy as np
3
4  S_A, S_B = y_A.sum(), y_B.sum()
5  # draw posterior samples (ps)
6  ps_A = beta(1 + S_A, 1 + (N - S_A), size=2000)
7  ps_B = beta(1 + S_B, 1 + (N - S_B), size=2000)
8
9  # posterior difference and 95% QBI
10 ps_diff = ps_A - ps_B
11 bayes_diff = np.percentile(ps_diff, [2.5, 97.5])
12
13 # posterior odds ratio and 95% QBI
14 ps_or = (ps_A / (1 - ps_A)) / (ps_B / (1 - ps_B))
15 bayes_or = np.percentile(ps_or, [2.5, 97.5])
```

**Remark 1** (**Bayesian model comparison**). *A key benefit of the Bayesian approach to model comparison is that we can easily compute probabilities that one model outperforms another. Given posterior samples, $\theta_m^{(k)} \sim p(\theta_m \mid y_{m,1:N})$, for models $m \in \{A, B\}$, we can calculate:*

$$\mathbb{P}(\theta_A > \theta_B \mid y_{A;1:N}, y_{B;1:N}) = \frac{1}{K} \sum_{k=1}^{K} \mathbb{1}[\theta_A^{(k)} > \theta_B^{(k)}].$$

*We cannot make such probabilistic statements in frequentist inference since the parameters are treated as fixed (but unknown) constants and probabilities only refer to hypothetical repetitions of the data.*

### 3.4. Failure of CLT-Based Confidence Intervals in Paired Model Comparison Settings

It is important to also consider model comparison in the paired setting, where two models have been evaluated on the *same* set of questions. A natural approach is to construct a CLT-based interval on $\theta_A - \theta_B$ using Eq. 4 which directly takes into account the paired structure of the data. Paired intervals are advantageous because evaluating both models on the same questions lets common question-specific effects

cancel out, which reduces variance. This leads to a more precise comparison if the outcomes are positively correlated, though negative correlation might increase variance.

To simulate paired evals data, we first sample probabilities of success for each model, alongside a correlation $\rho = 2\hat{\rho} - 1$:

$$\theta_A, \theta_B \sim \text{Beta}(1, 1), \quad \hat{\rho} \sim \text{Beta}(4, 2).$$

This formulation encourages positive correlations, which is expected in the context of LM evals: when models are evaluated on the same questions, factors such as question difficulty tend to affect both models similarly, leading to positively correlated performance (more details in Appendix C).

Next, we sample $N$ points from a bivariate Gaussian

$$(a_1, b_1), \ldots, (a_N, b_N) \stackrel{\text{IID}}{\sim} \mathcal{N}\left(\begin{pmatrix} \Phi^{-1}(\theta_A) \\ \Phi^{-1}(\theta_B) \end{pmatrix}, \begin{pmatrix} 1 & \rho \\ \rho & 1 \end{pmatrix}\right),$$

where $\Phi(\cdot)$ is the standard univariate Gaussian CDF. This parameterisation of the Gaussian covariance ensures that each 2D point has unit variances and correlation $\rho$. Meanwhile, the choice of the Gaussian's mean ensures that if we obtain binary eval outcomes for model A and model B by considering the sign of $a_i$ and $b_i$ respectively:

$$y_{A;i} = \mathbb{1}[a_i > 0], \quad y_{B;i} = \mathbb{1}[b_i > 0],$$

and the marginal probabilities of success for both models are as desired, that is, $\mathbb{P}(y_{A;i}) = \theta_A$ and $\mathbb{P}(y_{B;i}) = \theta_B$.

Much like in the clustered setting (§ 3.2), we perform Bayesian inference on the posterior distribution of $\theta_A - \theta_B$ using importance sampling, drawing $K = 10,000$ samples from the prior as the proposal distribution. Details and code for this can be found in Appendix E.2. We also present an 'unpaired Bayes' method in which we construct credible intervals on $\theta_A - \theta_B$ by sampling from the posteriors obtained for each of $\theta_A$ and $\theta_B$ separately as in § 3.3.

We see in Fig. 5 that all non-Bayesian methods severely underperform when it comes to achieving nominal coverage for small $N$. Moreover, the advantages of Bayesian

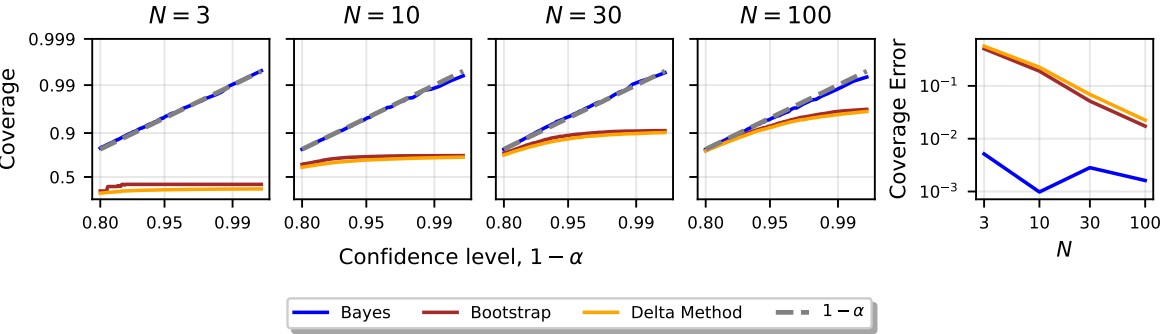

Figure 6: $F_1$**-score error bars**. Coverage vs. confidence level for Bayesian and bootstrap intervals. The CLT is not directly applicable due to the non-linearity of the $F_1$-score, so we use the the delta method to construct a CLT-based CI.

inference for enabling direct probabilistic comparison, as discussed in Remark 1, apply equally well in this setting. We would recommend using the paired Bayes method as it can account for correlations and thus produce narrower intervals. The unpaired Bayes is also a reasonable and easier to implement alternative (see Appendix I.4 for ablations).

### 3.5. Failure of CLT-Based Confidence Intervals When Metrics Are Not Simple Averages

Many metrics for LLM evals cannot be represented as simple averages of IID variables, in which case the CLT cannot be used to construct confidence intervals at all. We already saw this issue with the odds ratio in § 3.3, but the problem extends to many other widely used metrics. Indeed, some recent work, e.g. the Llama 3 report, acknowledges this limitation and omits reporting confidence intervals for metrics that are not simple averages (see p.29 in Dubey et al., 2024).

For many tasks (e.g. retrieval), it makes sense to not only track whether a model gave a correct or incorrect response, but also whether that response was a true positive (TP), true negative (TN), false positive (FP), or a false negative (FN). These counts form a 2×2 contingency table known as a confusion matrix. The outcome $y_i$ can therefore be viewed as a draw from a Categorical distribution with some parameter $\boldsymbol{\theta} := (\theta_{\text{TP}}, \theta_{\text{FP}}, \theta_{\text{FN}}, \theta_{\text{TN}})$, and the total counts in each category $N_{\text{conf}} := (N_{\text{TP}}, N_{\text{FP}}, N_{\text{FN}}, N_{\text{TN}})$ is a draw from a Multinomial$(N, \boldsymbol{\theta})$. To simulate an evaluation dataset we sample ground truth parameters from a uniform Dirichlet prior, which is conjugate to the categorical likelihood:

$$\boldsymbol{\theta} \sim \text{Dirichlet}(1, 1, 1, 1),$$
$$y_i \sim \text{Categorical}(\boldsymbol{\theta}), \qquad (9)$$
$$\boldsymbol{\theta} \,|\, y_{1:N} \sim \text{Dirichlet}(1 + N_{\text{conf}}).$$

Many metrics derived from the confusion matrix, e.g. $F_\beta$-scores, MCC or G-score, are non-linear in $\boldsymbol{\theta}$, so the CLT is not applicable (Caelen, 2017). Under standard regularity conditions, the delta method (Oehlert, 1992) can be used to in conjunction with the CLT to approximate the sampling

distribution of smooth non-linear functions of estimators via a first-order Taylor expansion (see Appendix A for details).

As an example, consider the $F_1$ score, which is the harmonic mean of precision and recall:

$$F_1 = 2 \, \frac{\text{precision} \cdot \text{recall}}{\text{precision} + \text{recall}}, \qquad (10)$$

where $\text{precision} = \frac{N_{\text{TP}}}{N_{\text{TP}} + N_{\text{FP}}}$ and $\text{recall} = \frac{N_{\text{TP}}}{N_{\text{TP}} + N_{\text{FN}}}$.

Our empirical evaluation is therefore focused only on comparing Bayesian credible intervals based on the model from Eq. 9 against the bootstrap, with results presented in Fig. 6. The Bayesian intervals closely track the nominal coverage, while the bootstrap ones systematically under-cover. We therefore recommend using Bayesian intervals in practice. The next code snippet demonstrates how to implement this approach. To show robustness to the choice of interval, we also include highest density intervals (HDIs), with results shown in Fig. 39 in the Appendix.

```
Snippet 3: Bayesian credible interval for F₁ score
1  from numpy.random import dirichlet
2  from arviz import hdi
3
4  # confusion_arr is np.array([N_TP,N_FP,N_FN,N_TN])
5  ps = dirichlet(confusion_arr + 1, 2000)
6  f1_samples = calculate_f1(ps) # implements Eq.10
7
8  # 95% HDI and QBI
9  bayes_hdi = hdi(f1_samples, hdi_prob=0.95)
10 bayes_qbi = np.percentile(f1_samples, [2.5, 97.5])
```

## 4. Alternative Views

It may be argued that CLT-based methods are usually sufficient in practice when their assumptions are satisfied. We do not disagree. However, we argue that it is safer to use the more robust strategies laid out in this paper, which are just as easy to apply (as demonstrated throughout), perform no worse for large $N$ and perform substantially better in the increasingly common small-$N$ setting. This is especially important since knowing whether a certain $N$ is "large enough"

Table 1: **Summary of interval methods and their key properties**. *Coverage* describes whether a method is able to provide at least the desired nominal coverage in small-sample settings. *Efficiency* describes how tight and precise the resulting intervals are given the nominal coverage (e.g., CLT-based intervals can be invalid or too wide). The *computational cost* of all methods is negligible compared to the cost of running the LLM evals, though we indicate their relative costs to facilitate comparison across methods.

| | Coverage small $N$ | Efficiency small $N$ | Computational cost | Easy to implement |
|---|---|---|---|---|
| CLT | ✗ | ✗ | Very low | Yes |
| CLT-based variants (e.g. Delta method) | ✗ | ✗ | Very low | Moderate |
| Custom frequentist (e.g. Wilson) | ✓ | ✓ | Very low | Moderate |
| Bootstrap | ✗ | ✗ | Low | Moderate |
| Bayes (conjugate) | ✓ | ✓ | Very low | Yes |
| Bayes (importance sampling) | ✓ | ✓ | Low | Moderate |

for the CLT to hold—a vague and unhelpful framing of the problem—would be extremely context-dependent and difficult to determine *a priori*. A key reason we chose to word the title of this paper as we did is to avoid implying the existence of some hard-valued $N^*$ below which the CLT is always invalid and unjustifiable, and above which it is always sound and reliable.

We argued that Bayesian credible intervals can be useful and flexible alternatives to CLT-based confidence intervals, particularly in settings where other frequentist methods are either too complicated or non-existent. Two common criticisms of Bayesian methods are that they are sensitive to the choice of a prior and that they can be computationally expensive. It is worth exploring these points in some detail with respect to the Bayesian methods discussed above.

Throughout this paper we use broad, non-informative, uniform priors over model performance to ensure that error bars are determined only by the benchmark results, and not by strong prior beliefs about how we might expect a certain model to perform. Whilst incorporating such information into an *informative* or *subjective* prior can lead to tighter error bars, it comes with important caveats. First, achieving optimal coverage and efficiency (i.e. producing the smallest interval width) requires knowing the true underlying prior information (Severini, 1993). In practice, this information is typically unavailable or unreliable. Second, subjective priors are often viewed as controversial: the choice of a specific prior can be arbitrary or unjustified, which in turn will introduce unwanted biases (Efron, 1986; Gelman, 2008).

Nonetheless, considering the impact of priors is critical in any Bayesian procedure. In Appendix I we explore the effect of prior mismatch across each of the experimental settings (§3.1–3.5). In these ablations, we continue to use our non-informative uniform prior for inference, but consider scenarios where we *could* incorporate additional prior information. For example, Fig. 18 and Fig. 19 consider the cases where a model is expected to perform well or badly,

respectively. We find that Bayesian coverage performance generally does not fall below that of CLT-based methods and often still outperforms them, especially for small $N$.

Finally, the added computational cost of Bayesian inference is negligible compared to the overall cost of benchmark construction and running the LLM evals themselves (see Appendix F), making it a worthwhile trade-off for improved accuracy in uncertainty quantification. In Table 1 we summarise these comparisons in terms of interval quality, computational cost and ease of implementation.

## 5. Conclusion

In this position paper, we argued against using the CLT to construct confidence intervals for LLM evals because the assumptions—a *large* number of *independent* samples—are rarely satisfied. LLM evals often have highly structured correlations among questions, correlated model outputs, and rely on increasingly smaller, specialized benchmarks. The CLT also does not apply to common metrics like $F$-scores that are not simple averages of IID variables. Of the alternatives that we examined, we found that boostrap intervals also perform poorly, while more appropriate frequentist methods and Bayesian credible intervals are much more reliable. We provided examples and code demonstrating how easy it is to implement these methods, and we recommend adopting them as standard practice for modern LLM evaluations.

## Acknowledgements

Sam Bowyer is supported by the UKRI EPSRC via the COMPASS CDT at the University of Bristol (EPS0235691). This work was possible thanks to the computational facilities of the University of Bristol's Advanced Computing Research Centre—http://www.bris.ac.uk/acrc/. We would like to thank Dr. Stewart for funding compute resources used in this project. DRI would like to thank Gregorio Benincasa for a helpful discussion.

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

# A. Frequentist Confidence Intervals

## A.1. Confidence Intervals Based on the t-Distribution

When the data is normally distributed with unknown mean and variance, $X_1, \ldots, X_N \overset{\text{IID}}{\sim} \mathcal{N}(\mu, \sigma^2)$, the Student's t-distribution provides an exact *finite* sample solution for confidence intervals (and hypothesis tests):

$$\text{CI}_{1-\alpha}(\mu) = \hat{\mu} \pm t_{\alpha/2, \nu} \, \text{SE}(\hat{\mu}), \tag{11}$$

where $t_{\alpha/2, \nu}$ is the $\alpha/2$-th quantile of the Student's t-distribution with $\nu = N - 1$ degrees of freedom. When $N \approx 30$ (or above), the t-distribution is close to the standard normal (e.g. $t_{0.025, 29} = 2.045$ vs $z_{0.025} = 1.960$).

Confidence intervals based on Eq. 11 are often used even when the data is not normally distributed. In this case, the following assumptions are required for exactness:

- The sample mean, $\hat{\mu}$, is approximately normally distributed (which by the CLT holds for large enough sample sizes).

- The quantity $\frac{S^2(N-1)}{\sigma^2}$ follows a Chi-Squared distribution with $N-1$ degrees of freedom, $\chi^2(N-1)$, that is **independent** of the sample mean, $\hat{\mu}$.

Recall that in § 2, we relied on the Slutsky's theorem to deal with the unknown variance. For smaller sample sizes, if the two assumptions above are approximately satisfied, the t-based intervals can have better properties than z-based ones. However, for the binary LLM evals we consider here, the variance $S^2$ is not independent of the sample mean. Nevertheless, in Fig. 7 we show the properties of a t-based interval for the independent model comparison setting, where we construct intervals on the difference in means $\theta_A - \theta_B$.

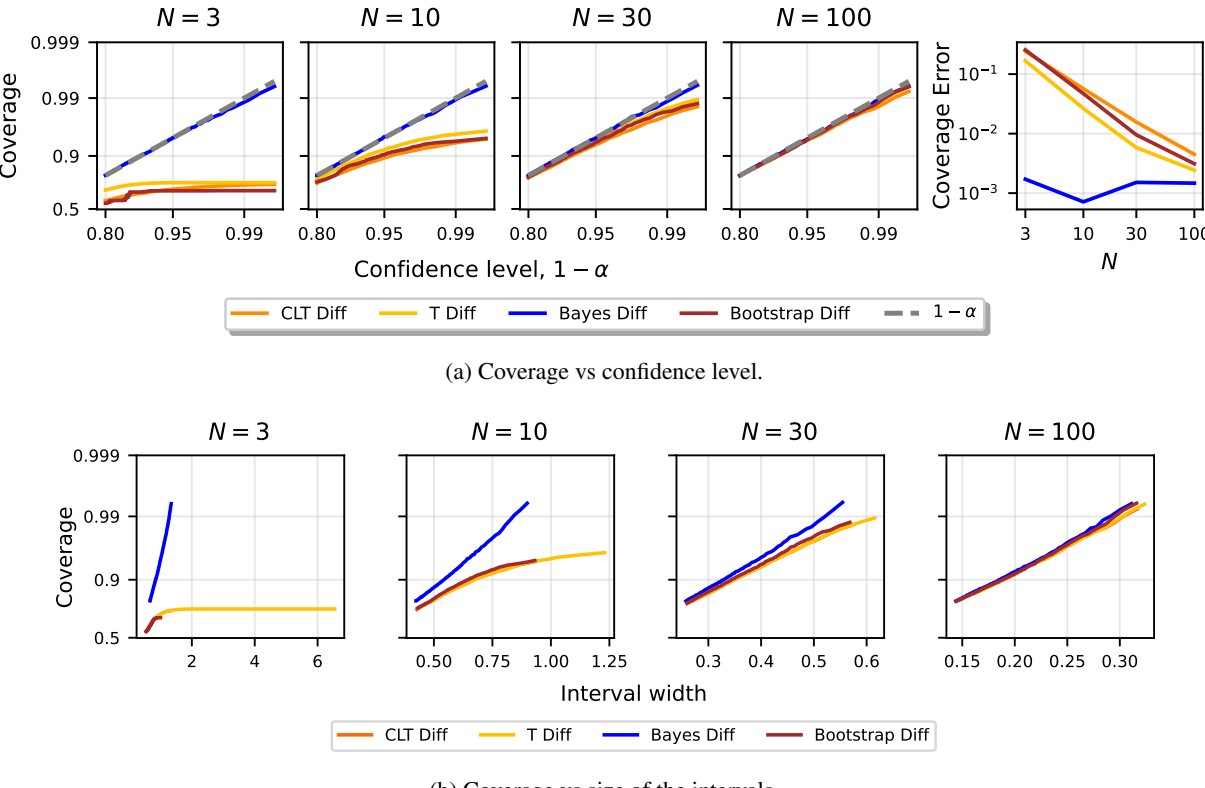

(a) Coverage vs confidence level.

(b) Coverage vs size of the intervals.

Figure 7: **Independent model comparison setting**. Intervals for the difference in means $\theta_A - \theta_B$.

Fig. 7a shows more favourable coverage properties compared to the CLT-based intervals. However, these intervals are

extremely wide (Fig. 7b) with width exceeding 1 for small $N$. Given that the interval $[0, 1]$ achieves 100% coverage for any $N$, the t-based intervals are clearly not useful.

## A.2. Wilson Score Intervals

The Wilson score (WS) interval (Wilson, 1927) is applicable when constructing CIs for a single model's accuracy. It has improved coverage properties over the CLT approximation, particularly with small sample sizes.

Given Bernoulli data $y_i \overset{\text{IID}}{\sim} \text{Bernoulli}(\theta)$, $i = 1, \ldots, N$, the WS interval is:

$$\text{CI}_{1-\alpha,\text{Wilson}}(\theta) = \frac{\hat{\theta} + \frac{z_{\alpha/2}^2}{2N}}{1 + \frac{z_{\alpha/2}^2}{N}} \pm \frac{\frac{z_{\alpha/2}}{2N}}{1 + \frac{z_{\alpha/2}^2}{N}} \sqrt{4N\hat{\theta}(1 - \hat{\theta}) + z_{\alpha/2}^2} \tag{12}$$

where $z_{\alpha/2}$ is the $\alpha/2$-th quantile of the standard normal distribution. (Note that the centre of the interval is no longer the sample mean $\hat{\theta}$—this helps to avoid the interval from collapsing to zero-width or extending past $[0, 1]$.)

To arrive at this, we first take a normal approximation of the binomial with the sample standard deviation given by $\sqrt{\theta(1 - \theta)/N}$:

$$z_{\alpha/2} \approx \frac{\theta - \hat{\theta}}{\sqrt{\theta(1 - \theta)/N}}. \tag{13}$$

Rearranging this we get a quadratic in $\theta$

$$\theta(1 - \theta)z_{\alpha/2}^2 = (\theta - \hat{\theta})^2 \tag{14}$$

$$0 = (N + z_{\alpha/2}^2)\theta^2 - (2N\hat{\theta} + z_{\alpha/2}^2)\theta + N\hat{\theta}^2 \tag{15}$$

which we can solve using the standard quadratic formula to find the upper and lower values of $\theta$ for the $1 - \alpha$ confidence interval

$$p = \frac{(2N\hat{\theta} + z_{\alpha/2}^2) \pm \sqrt{(2N\hat{\theta} + z_{\alpha/2}^2)^2 - 4(N + z_{\alpha/2}^2)(N\hat{\theta}^2)}}{2(N + z_{\alpha/2}^2)} \tag{16}$$

$$= \frac{2N\hat{\theta} + z_{\alpha/2}^2}{2(N + z_{\alpha/2}^2)} \pm \frac{\sqrt{4N^2\hat{\theta}^2 + 4N\hat{\theta}z_{\alpha/2}^2 + z_{\alpha/2}^4 - 4N^2\hat{\theta}^2 - 4N\hat{\theta}^2 z_{\alpha/2}^2}}{2(N + z_{\alpha/2}^2)} \tag{17}$$

$$= \frac{2N\hat{\theta} + z_{\alpha/2}^2}{2(N + z_{\alpha/2}^2)} \pm \frac{z_{\alpha/2}\sqrt{4N\hat{\theta} - 4N\hat{\theta}^2 + z_{\alpha/2}^2}}{2(N + z_{\alpha/2}^2)} \tag{18}$$

$$= \frac{2N\hat{\theta} + z_{\alpha/2}^2}{2(N + z_{\alpha/2}^2)} \pm \frac{z_{\alpha/2}\sqrt{4N\hat{\theta}(1 - \hat{\theta}) + z_{\alpha/2}^2}}{2(N + z_{\alpha/2}^2)} \tag{19}$$

$$= \frac{\hat{\theta} + \frac{z_{\alpha/2}^2}{2N}}{1 + \frac{z_{\alpha/2}^2}{N}} \pm \frac{\frac{z_{\alpha/2}}{2N}}{1 + \frac{z_{\alpha/2}^2}{N}} \sqrt{4N\hat{\theta}(1 - \hat{\theta}) + z_{\alpha/2}^2}. \tag{20}$$

Thus arriving at Eq. 12.

## A.3. Clopper-Pearson Intervals

Similarly to WS, the Clopper-Pearson (CP) interval (Clopper & Pearson, 1934) is applicable when constructing CIs for a single model's accuracy. It is an *exact* method based on the cumulative binomial distribution, albeit often yielding conservative (wider) intervals.

Given Bernoulli data $y_i \overset{\text{IID}}{\sim} \text{Bernoulli}(\theta)$, $i = 1, \ldots, N$, the CP interval is defined as containing all values of $\theta$ for which a two-sided binomial hypothesis test with significance level $\alpha$ does *not* reject the null hypothesis $H_0 : \theta = \hat{\theta}$ in favour of the

alternative $H_1 : \theta \neq \hat{\theta}$, resulting in a coverage that is guaranteed to be at least $1 - \alpha$. We may write the CP interval as

$$\mathrm{CI}_{1-\alpha,\mathrm{CP}}(\theta) = [\theta_{\mathrm{lower}}, \theta_{\mathrm{upper}}], \tag{21}$$

where $\theta_{\mathrm{lower}}$ and $\theta_{\mathrm{upper}}$ are such that (denoting $\bar{y} = \frac{1}{N}\sum_{i=1}^N y_i$)

$$\sum_{k=N\bar{y}}^N \binom{N}{k} \theta_{\mathrm{lower}}^k (1 - \theta_{\mathrm{lower}})^{n-k} = \frac{\alpha}{2} \tag{22}$$

$$\sum_{k=0}^{N\bar{y}} \binom{N}{k} \theta_{\mathrm{upper}}^k (1 - \theta_{\mathrm{upper}})^{n-k} = \frac{\alpha}{2}. \tag{23}$$

It can be shown (Thulin, 2014) that the values of $\theta_{\mathrm{lower}}$ and $\theta_{\mathrm{upper}}$ are given by

$$\theta_{\mathrm{lower}} = B\left(\frac{\alpha}{2}, \sum_{i=1}^N y_i, 1 + \sum_{i=1}^N (1 - y_i)\right) \tag{24}$$

$$\theta_{\mathrm{upper}} = B\left(1 - \frac{\alpha}{2}, 1 + \sum_{i=1}^N y_i, \sum_{i=1}^N (1 - y_i)\right) \tag{25}$$

where $B(\alpha, a, b)$ is the $\alpha$-th quantile of the $\mathrm{Beta}(a, b)$ distribution.

### A.4. Delta Method

As discussed in § 3.5, in some settings the metric we are interested in is not $\theta$ itself, but some value $g(\theta)$ (such as the $F_1$-score). The usual way to estimate $g(\theta)$ is to use the *plug-in estimator* $g(\hat{\theta})$, where $\hat{\theta}$ is the maximum likelihood estimate of $\theta$. In this case, the CLT cannot be directly applied to obtain confidence intervals for $g(\theta)$.

Under standard regularity conditions, the delta method (Oehlert, 1992) can be used in conjunction with the CLT to approximate the sampling distribution of such metrics. By the (multivariate) CLT we have

$$\sqrt{n}(\hat{\theta} - \theta^\star) \xrightarrow{d} \mathcal{N}(0, \Sigma) \tag{26}$$

Let $g : \theta \mapsto g(\theta)$ be a differentiable function. Then

$$\sqrt{n}(g(\hat{\theta}) - g(\theta)) \xrightarrow{d} \mathcal{N}(0, \nabla g(\theta)^\top \Sigma \nabla g(\theta)). \tag{27}$$

This allows us to construct an approximate $(1 - \alpha)$ confidence interval for $g(\theta)$ as follows:

$$\mathrm{CI}_{1-\alpha}(g(\theta)) = g(\hat{\theta}) \pm z_{\alpha/2} \sqrt{\frac{1}{n} \nabla g(\hat{\theta})^\top \Sigma \nabla g(\hat{\theta})}. \tag{28}$$

As before, we can rely on Slutsky's theorem and use the sample covariance $\hat{\Sigma}$ if the population covariance is unknown.

### A.5. Delta method for the $F_1$ score

Let $\boldsymbol{\theta} = (\theta_{\mathrm{TP}}, \theta_{\mathrm{FP}}, \theta_{\mathrm{FN}}, \theta_{\mathrm{TN}})^\top$. The $F_1$ score can be written as

$$g(\boldsymbol{\theta}) = F_1 = \frac{2\theta_{\mathrm{TP}}}{2\theta_{\mathrm{TP}} + \theta_{\mathrm{FP}} + \theta_{\mathrm{FN}}} = \frac{2\theta_{\mathrm{TP}}}{d} \quad d := 2\theta_{\mathrm{TP}} + \theta_{\mathrm{FP}} + \theta_{\mathrm{FN}}. \tag{29}$$

Then we have for the gradient $\nabla g(\boldsymbol{\theta})$:

$$\nabla g(\boldsymbol{\theta}) = \begin{pmatrix} \dfrac{\partial g}{\partial \theta_{\mathrm{TP}}} \\ \dfrac{\partial g}{\partial \theta_{\mathrm{FP}}} \\ \dfrac{\partial g}{\partial \theta_{\mathrm{FN}}} \\ \dfrac{\partial g}{\partial \theta_{\mathrm{TN}}} \end{pmatrix} = \begin{pmatrix} \dfrac{2(\theta_{\mathrm{FP}} + \theta_{\mathrm{FN}})}{d^2} \\ -\dfrac{2\theta_{\mathrm{TP}}}{d^2} \\ -\dfrac{2\theta_{\mathrm{TP}}}{d^2} \\ 0 \end{pmatrix} = \begin{pmatrix} \dfrac{2(1 - F_1)}{d} \\ -\dfrac{F_1}{d} \\ -\dfrac{F_1}{d} \\ 0 \end{pmatrix} \tag{30}$$

For the covariance of $\boldsymbol{\theta}$ we have:

$$\mathrm{Cov}(\boldsymbol{\theta}) = \frac{1}{N}\left(\mathrm{diag}(\boldsymbol{\theta}) - \boldsymbol{\theta}\boldsymbol{\theta}^\top\right) \tag{31}$$

The sample estimates of $\boldsymbol{\theta}$, $\mathrm{Cov}(\boldsymbol{\theta})$ and $F_1$ are:

$$\hat{\boldsymbol{\theta}} = \left(\frac{N_{\mathrm{TP}}}{N}, \frac{N_{\mathrm{FP}}}{N}, \frac{N_{\mathrm{FN}}}{N}, \frac{N_{\mathrm{TN}}}{N}\right)^\top \tag{32}$$

$$\mathrm{Cov}(\boldsymbol{\theta}) = \frac{1}{N}\left(\mathrm{diag}(\hat{\boldsymbol{\theta}}) - \hat{\boldsymbol{\theta}}\,\hat{\boldsymbol{\theta}}^\top\right) \tag{33}$$

$$\hat{F}_1 = \frac{2N_{\mathrm{TP}}}{2N_{\mathrm{TP}} + N_{\mathrm{FP}} + N_{\mathrm{FN}}} \tag{34}$$

which we use to get the sample variance of $g(\boldsymbol{\theta}) = F_1$ as per Eq. 27:

$$\mathrm{Var}(F_1) = \nabla g(\boldsymbol{\theta})^\top \Sigma \nabla g(\boldsymbol{\theta}) \tag{35}$$

$$\approx \nabla g(\hat{\boldsymbol{\theta}})^\top \mathrm{Cov}(\hat{\boldsymbol{\theta}}) \nabla g(\hat{\boldsymbol{\theta}}) \tag{36}$$

$$= \frac{1}{N} \nabla g(\hat{\boldsymbol{\theta}})^\top \left[\mathrm{diag}(\hat{\boldsymbol{\theta}}) - \hat{\boldsymbol{\theta}}\,\hat{\boldsymbol{\theta}}^\top\right] \nabla g(\hat{\boldsymbol{\theta}}) \tag{37}$$

$$= \frac{\hat{F}_1\,(1 - \hat{F}_1)\,(2 - \hat{F}_1)}{N\,d} \tag{38}$$

$$= \frac{\hat{F}_1\,(1 - \hat{F}_1)\,(2 - \hat{F}_1)}{2\,N_{\mathrm{TP}} + N_{\mathrm{FP}} + N_{\mathrm{FN}}} \tag{39}$$

Thus the standard error is

$$\mathrm{SE}(\hat{F}_1) = \sqrt{\frac{\hat{F}_1(1 - \hat{F}_1)(2 - F_1)}{2\,N_{\mathrm{TP}} + N_{\mathrm{FP}} + N_{\mathrm{FN}}}}, \tag{40}$$

and the approximate two-sided $100(1 - \alpha)\%$ confidence interval is

$$\mathrm{CI}_{1-\alpha}(F_1) = \hat{F}_1 \pm z_{1-\alpha/2}\sqrt{\frac{F_1(1 - F_1)(2 - F_1)}{2\,N_{\mathrm{TP}} + N_{\mathrm{FP}} + N_{\mathrm{FN}}}}. \tag{41}$$

# B. Real-World Eval Error Bars

## B.1. Full LangChain Eval Error Bars

Here we present the error bars on all LLMs present in the Langchain dataset in Fig. 8 for which we could find response data publicly on all $N = 20$ questions[1]. The evals represent model success on Langchain's 26-tool typewriter task, in which LLM agents must spell out strings of letters by using 26 tools which each represent a letter of the alphabet. Below, we clarify the more specific names of some of the LLMs in the figure, with the GPT models marked by a [†] being the ones shown in Fig. 1.

- GPT-4[†]: `gpt-4-1106-preview`

- Mixtral-8-7B: `mixtral-8x7b-instruct`

- GPT-3.5[†]: `gpt-3.5-turbo-0613-openai`

- GPT-4[‡]: `gpt-4-0613`

- GPT-3.5[‡]: `gpt-3.5-turbo-1106`

- Llama-2-70B: `llama-v2-70b-chat`

- Mistral-7B: `mistral-7b-instruct`

- Llama-2-13B: `llama-v2-13b-chat`

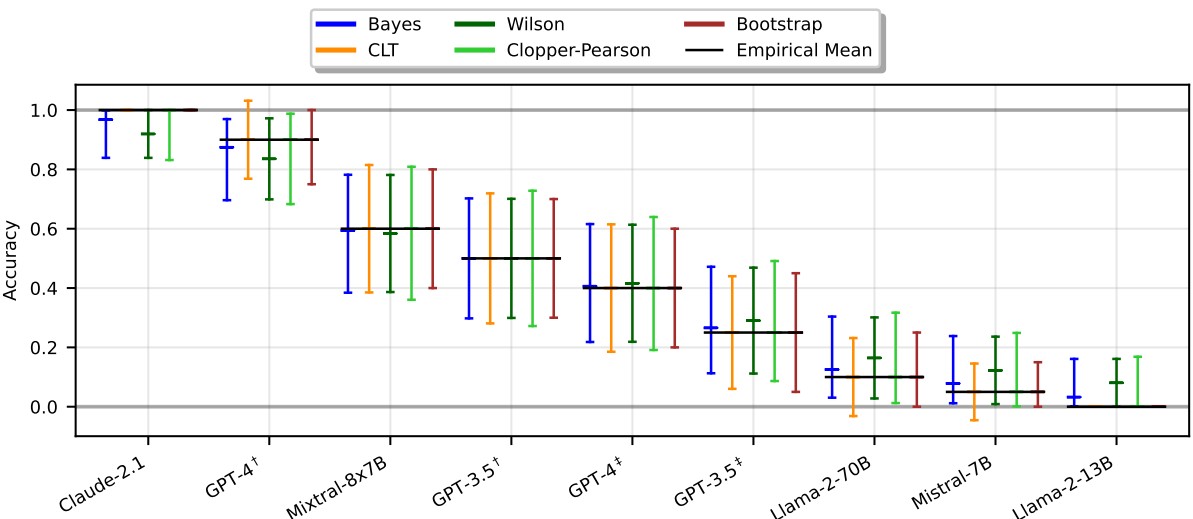

Figure 8: **Error bars on Langchain Tool-use Benchmark**. Extended version of Fig. 1. The benchmark consists of $N = 20$ questions and we show 95% confidence/credible intervals for the model accuracy, with the empirical mean shown in black.

---

[1]The raw evals data can be found along with code to reproduce all experiments in this paper at `https://github.com/sambowyer/no_clt_paper`.

**B.2. Math Arena AIME 2025 II Error Bars**

In Fig. 9 we present error bars for all models on the Math Arena AIME 2025 II benchmark (Balunović et al., 2025; AIME II, 2025). We take each model's first attempt at each of the $N = 15$ questions and compute 95% confidence/credible intervals using the methods laid out in § 3.1. Note that we again observe CLT-based error bars collapsing to zero-width (as do bootstrap error bars) and extending past $[0, 1]$.

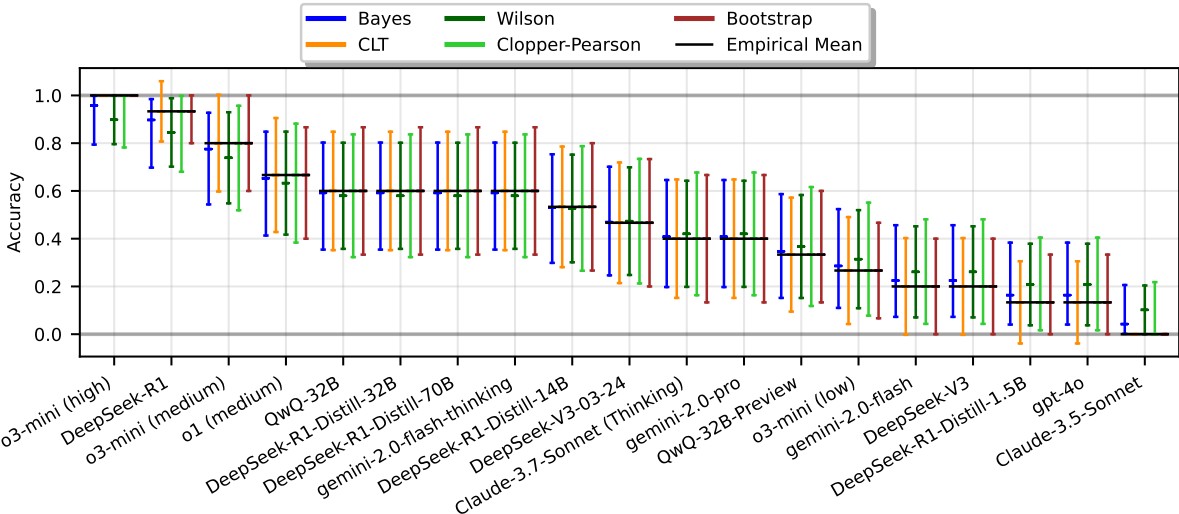

Figure 9: Error bars at 95% confidence level computed for all models available on the Math Arena AIME 2025 II Benchmark ($N = 15$).

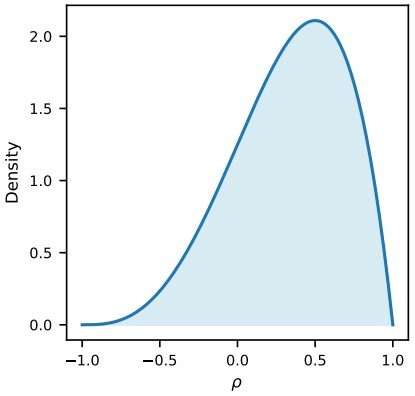
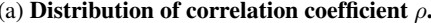

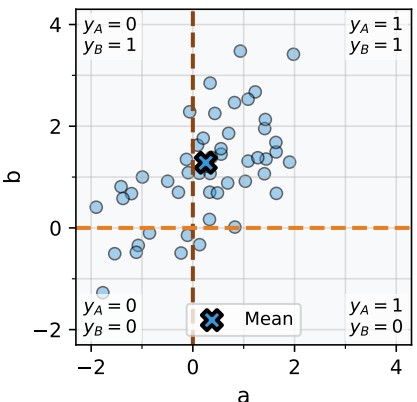

(a) **Distribution of correlation coefficient $\rho$.**

(b) **Paired model data generation illustration** with $\rho = 0.7$, $\theta_A = 0.6$, $\theta_B = 0.9$ and $N = 50$ samples.

Figure 10

## C. Simulating Correlated Paired-Model Eval Data

In § 3.4, we wanted to generate pairs of eval results for two LLMs, $A$ and $B$, such that the responses between the two models were correlated. In order to maintain control on the marginal values of $\theta_A$ and $\theta_B$, we sample these uniformly in the same way as in the rest of the paper:

$$\theta_A, \theta_B \overset{\text{IID}}{\sim} \text{Beta}(1,1) = \text{Uniform}[0,1]. \tag{42}$$

However, to induce correlation between evals we clearly can't just use these values as Bernoulli parameters independently. First, we sample a correlation coefficient $\rho = 2\hat{\rho} - 1$ where $\hat{\rho} \sim \text{Beta}(4,2)$, leading to the distribution of $\rho$ shown in Fig. 10a. The preference for positive correlations follows on from real-world intuition: we'd expect results from two LLMs to be positively correlated much more of the time than negatively correlated.

Next, we sample $N$ points, $\{(a_i, b_i)\}_{i=1}^{N}$, on a bivariate Gaussian with unit variances and correlation coefficient $\rho$:

$$(a_i, b_i) \overset{\text{IID}}{\sim} \mathcal{N}\left((\mu_A \quad \mu_B)^{\top}, \Sigma\right), \tag{43}$$

$$\mu_A = \Phi^{-1}(\theta_A), \tag{44}$$

$$\mu_B = \Phi^{-1}(\theta_B), \tag{45}$$

$$\Sigma = \begin{pmatrix} 1 & \rho \\ \rho & 1 \end{pmatrix}, \tag{46}$$

where $\Phi$ is the standard univariate Gaussian CDF.

Since the marginal distributions of $a_i$ and $b_i$ are standard Gaussians centred at $\mu_A$ and $\mu_B$ respectively, we have that $(a_i - \mu_A), (b_i - \mu_B) \sim \mathcal{N}(0,1)$ and therefore

$$\mathbb{P}(a_i > 0) = 1 - \mathbb{P}(a_i - \mu_A < -\mu_A) = 1 - \Phi(-\mu_A) = \Phi(\mu_A) = \theta_A, \tag{47}$$

$$\mathbb{P}(b_i > 0) = 1 - \mathbb{P}(b_i - \mu_B < -\mu_B) = 1 - \Phi(-\mu_B) = \Phi(\mu_B) = \theta_B. \tag{48}$$

Hence, if we assign binary eval outcomes for both LLMs according to the signs of $a_i$ and $b_i$ for $i = 1, \ldots, N$, we arrive at Bernoulli marginals, as desired:

$$y_{A;i} = \mathbb{1}[a_i > 0], \tag{49}$$

$$y_{B;i} = \mathbb{1}[b_i > 0]. \tag{50}$$

The choice of the covariance matrix $\Sigma$ ensures that the evals for the two LLMs are are correlated with the desired level of correlation $\rho$.

We illustrate this data generation procedure in Fig. 10b. We generate a paired eval dataset with a positive correlation of $\rho = 0.7$ between the two models, A and B. The success probabilities are $\theta_A = 0.6$ and $\theta_B = 0.9$. The points in the scatter plot are samples from a bivariate Gaussian distribution with mean $\left( \Phi^{-1}(\theta_A) \quad \Phi^{-1}(\theta_B) \right)^{\top} = (0.25 \quad 1.28)^{\top}$ (denoted with a blue cross), unit variance and correlation $0.7$. The threshold lines at $a = 0$ (dashed orange horizontal line) and $b = 0$ (dashed brown vertical line) divide the space into four quadrants, each corresponding to a different combination of binary outcomes $y_A$ and $y_B$, as labelled in the corners.

See Appendix E.2 for details on how inference is done in this model via importance sampling.

## D. Pass@K Metric

In § 3.5 we examined $F_1$ scores as a metric more immediately amenable to Bayesian than frequentist analysis, but many other metrics would also fit in here too. For example, consider the common pass@K metric, which reports the proportion of runs in which a model gives the correct answer within its first $K$ (IID) generations.

In the Bayesian setting, we can infer a Bernoulli posterior over the probability of a single generation on a single question being correct for a certain model. This can then easily be used to compute error bars for the probability of *at least* 1 in $K$ (IID) generations being correct.

However, since we're really interested in a single number that measures the performance of a model on all the questions in a dataset, we'd need a hierarchical Bayesian model with a per-model latent variable, which captures the average probability of a single generation being correct on a single, randomly chosen question. This might look similar to the hierarchical clustered model from § 3.2, but with a hierarchy over models rather than tasks. This model could also be extended to have an hierarchy over both models and questions/tasks. It is unclear to us how best to construct a corresponding CLT-based/frequentist interval.

# E. Python Code for Importance Sampling

Here we provide simple code for performing the importance sampling mentioned in § 3.2 and § 3.4. For a review of importance sampling techniques, see (Tokdar & Kass, 2010).

In general, importance sampling for Bayesian inference works by drawing some $K$ samples, $\{\theta^{(k)}\}_{k=1}^{K}$, of a latent variable $\theta$ from some proposal distribution $Q$ and computing their importance weight as the ratio between the likelihood under the generative model $P(Y, \theta)$ and the proposal $Q(\theta)$,

$$w^{(k)} = \frac{P(Y, \theta)}{Q(\theta)} \tag{51}$$

where $Y$ is our data. Using Bayes' rule, we can rewrite this to show that if our proposal is simply the prior, $Q(\theta) = P(\theta)$, then the importance weights are directly given by the likelihood

$$w^{(k)} = \frac{P(Y, \theta^{(k)})}{Q(\theta^{(k)})} \tag{52}$$

$$= \frac{P(Y|\theta^{(k)})P(\theta^{(k)})}{Q(\theta^{(k)})} \tag{53}$$

$$= P(Y|\theta^{(k)}). \tag{54}$$

Then we can, for example, approximate the posterior expectation of the latent $\theta$ using normalised versions of the importance weights

$$\hat{w}^{(k)} = \frac{w^{(k)}}{\sum_{i=1}^{K} w^{(i)}} \tag{55}$$

since

$$\mathbb{E}_{\theta \sim P(\cdot|Y)}[\theta] = \int \theta P(\theta|Y) d\theta \tag{56}$$

$$= \int \theta \frac{P(Y|\theta)P(\theta)}{P(Y)} d\theta \tag{57}$$

$$= \int \theta \frac{P(Y|\theta)Q(\theta)}{P(Y)} d\theta \tag{58}$$

$$= \mathbb{E}_{\theta \sim Q(\cdot)} \left[ \theta \frac{P(Y|\theta)}{P(Y)} \right] \tag{59}$$

$$\approx \frac{1}{K} \sum_{k=1}^{K} \theta^{(k)} \hat{w}^{(k)}. \tag{60}$$

## E.1. Importance Sampling in the Clustered Setting

The generative model for the clustered setting is as follows:

$$d \sim \mathrm{Gamma}(1, 1) \tag{61}$$

$$\theta \sim \mathrm{Beta}(1, 1) \tag{62}$$

$$\theta_t \sim \mathrm{Beta}(d\theta, d(1 - \theta)) \tag{63}$$

$$y_{i,t} \sim \mathrm{Bernoulli}(\theta_t). \tag{64}$$

To perform Bayesian inference on $\theta$ we integrate out $\theta_t$:
$$Y_t \sim \mathrm{BetaBin}(N_t, d\theta, d(1 - \theta)). \tag{65}$$

where $Y_t = \sum_{i=1}^{N_t} y_{i,t}$ is the total number of correct answers in task $t$. As described in the main text, we use the prior as a proposal to obtain $K = 10,000$ samples $\{(\theta^{(k)}, d^{(k)})\}_{k=1}^{K}$ which have an associated importance weight:

$$w^{(k)} = \prod_{t=1}^{T} \mathrm{BetaBin}(Y_t; N_t, d^{(k)}\theta^{(k)}, d^{(k)}(1 - \theta^{(k)})). \tag{66}$$

We then resample the set $\{\theta^{(k)}\}_{k=1}^K$ with repeats using the weights $\{w^{(k)}\}_{k=1}^K$ and report credible intervals by taking the relevant percentiles of the resulting set of posterior samples.

---

**Snippet 4: Bayesian analysis for clustered evals**

```python
# S_t, N_t: np.arrays of length T with total
# sucesses & questions per task
import numpy as np
from scipy.stats import betabinom

# set number of samples, K
K = 10_000

# get K samples from the prior (with extra dimension for broadcasting over tasks)
thetas = np.random.beta(1,1, size=(K,1))
ds = np.random.gamma(1,1, size=(K,1))

# obtain weights via the likelihood (sum the per-task log-probs)
log_weights = betabinom(N_t, (ds*thetas), (ds*(1-thetas))).logpmf(S_t).sum(-1)

# normalise the weights
weights = np.exp(log_weights - log_weights.max())
weights /= weights.sum()

# obtain samples from the posterior
posterior = thetas[np.random.choice(K, size=K, replace=True, p=weights)]

# Bayesian credible interval
bayes_ci = np.percentile(posterior, [2.5, 97.5])
```

---

### E.2. Importance Sampling in the Paired Setting

In the paired setting we use the following generative model, as shown in Appendix C:

$$\theta_A \sim \text{Beta}(1,1) \tag{67}$$

$$\theta_B \sim \text{Beta}(1,1) \tag{68}$$

$$\hat{\rho} \sim \text{Beta}(4,2) \tag{69}$$

$$\rho = 2\hat{\rho} - 1 \tag{70}$$

$$(a_i, b_i) \overset{\text{IID}}{\sim} \mathcal{N}(\mu, \Sigma), \tag{71}$$

$$y_{A;i} = \mathbb{1}[a_i > 0], \tag{72}$$

$$y_{B;i} = \mathbb{1}[b_i > 0], \tag{73}$$

for $i = 1, \ldots, N$, such that

$$\mu = \begin{pmatrix} \Phi^{-1}(\theta_A) & \Phi^{-1}(\theta_B) \end{pmatrix}^T \tag{74}$$

$$\Sigma = \begin{pmatrix} 1 & \rho \\ \rho & 1 \end{pmatrix}, \tag{75}$$

where $\Phi$ is the standard univariate Gaussian CDF.

To perform Bayesian inference on $\theta_A - \theta_B$, we again use importance sampling where we draw $K = 10,000$ samples from the prior as a proposal to obtain $\{(\theta_A^{(k)}, \theta_B^{(k)}, \rho^{(k)})\}_{k=1}^K$. Then we calculate the importance weights as

$$w^{(k)} = \prod_{i=1}^N p(y_{A;i}, y_{B;i}|\theta_A^{(k)}, \theta_B^{(k)}, \rho^{(k)}). \tag{76}$$

In order to calculate this quantity, we break the problem into four cases, based on the possible combinations of success/failure for model A and model B. For a single question $i \in [N]$, we have

$$p(y_{A;i}, y_{B;i}|\theta_A^{(k)}, \theta_B^{(k)}, \rho^{(k)}) = \begin{cases} \theta_{AB}^{(k)} := \mathbb{P}(a_i, b_i > 0) & \text{if } a_i = b_i = 1, \\ \theta_{AB^\perp}^{(k)} := \mathbb{P}(a_i, > 0, b_i < 0) & \text{if } a_i = 1 \text{ and } b_i = 0 \\ \theta_{A^\perp B}^{(k)} := \mathbb{P}(a_i, < 0, b_i > 0) & \text{if } a_i = 0 \text{ and } b_i = 1 \\ \theta_{A^\perp B^\perp}^{(k)} := \mathbb{P}(a_i, b_i < 0) & \text{if } a_i = b_i = 0. \end{cases} \tag{77}$$

If we can figure out the values of $\theta_{AB}^{(k)}, \theta_{AB\perp}^{(k)}, \theta_{A\perp B}^{(k)}$, and $\theta_{A\perp B\perp}^{(k)}$, then we just need the number of occurrences of each combination in the data in order to evaluate Eq. 76:

$$w^{(k)} = \prod_{i=1}^{N} p(y_{A;i}, y_{B;i} | \theta_A^{(k)}, \theta_B^{(k)}, \rho^{(k)}) \tag{78}$$

$$= (\theta_{AB}^{(k)})^S (\theta_{AB\perp}^{(k)})^T (\theta_{A\perp B}^{(k)})^U (\theta_{A\perp B\perp}^{(k)})^V, \tag{79}$$

where

$$S = \sum_{i=1}^{N} y_{A;i} y_{B;i}, \tag{80}$$

$$T = \sum_{i=1}^{N} y_{A;i}(1 - y_{B;i}), \tag{81}$$

$$U = \sum_{i=1}^{N} (1 - y_{A;i}) y_{B;i}, \tag{82}$$

$$V = \sum_{i=1}^{N} (1 - y_{A;i})(1 - y_{B;i}). \tag{83}$$

Now, note that

$$\theta_{A\perp B\perp}^{(k)} = \mathbb{P}(a_i^{(k)}, b_i^{(k)} < 0) = \Phi_2(\mathbf{0}, ; \mu^{(k)}, \Sigma^{(k)}), \tag{84}$$

where $\Phi_2$ is the bivariate Gaussian CDF (parameterised by $\mu^{(k)}$ and $\Sigma^{(k)}$), which can be calculated numerically. Specifically, we adopt the simple approximation derived in Tsay & Ke (2023), an implementation of which can be found at https://github.com/sambowyer/bayes_evals.

Knowing $\theta_{A\perp B\perp}^{(k)}$ allows you to calculate the other three probabilities, since we know the following:

$$1 = \theta_{AB}^{(k)} + \theta_{AB\perp}^{(k)} + \theta_{A\perp B}^{(k)} + \theta_{A\perp B\perp}^{(k)}, \tag{85}$$

$$\theta_A^{(k)} = \theta_{AB}^{(k)} + \theta_{AB\perp}^{(k)}, \tag{86}$$

$$\theta_B^{(k)} = \theta_{AB}^{(k)} + \theta_{A\perp B}^{(k)}. \tag{87}$$

Thus, we can calculate the importance weights from Eq. 78 with the following relationships:

$$\theta_{AB}^{(k)} = \theta_{A\perp B\perp}^{(k)} + \theta_A^{(k)} + \theta_B^{(k)} - 1, \tag{88}$$

$$\theta_{AB\perp}^{(k)} = 1 - \theta_B^{(k)} - \theta_{A\perp B\perp}^{(k)}, \tag{89}$$

$$\theta_{A\perp B}^{(k)} = 1 - \theta_A^{(k)} - \theta_{A\perp B\perp}^{(k)}. \tag{90}$$

As in the previous section, App. E.1, we obtain a collection of $K$ posterior samples, $\{(\theta_A^{(k,\text{posterior})}, \theta_B^{(k,\text{posterior})})\}_{k=1}^{K}$, by resampling the set $\{(\theta_A^{(k)}, \theta_B^{(k)})\}_{k=1}^{K}$ with repeats using the weights $\{w^{(k)}\}_{k=1}^{K}$. To calculate our credible intervals on the value of $\theta_A - \theta_B$, we take the relevant percentiles from the set $\{\theta_A^{(k,\text{posterior})} - \theta_B^{(k,\text{posterior})}\}_{k=1}^{K}$.

Snippet 5: Bayesian analysis for paired evals

```python
1  # y_A, y_B: length N binary "eval" vectors
2  import numpy as np
3  from numpy.random import beta
4  from scipy.stats import norm
5  from binorm import binorm_cdf # 2D Gaussian CDF, defined elsewhere
6
7  # set number of samples, K
8  K = 10_000
9
10 # get K samples from the prior (with extra dimension for broadcasting over questions)
11 theta_As = beta(1,1, size=K)
12 theta_Bs = beta(1,1, size=K)
13 rhos     = 2*beta(4,2, size=K) - 1
14
15
16 # 2x2 contingency table (flattened)
17 S = (y_A * y_B).sum(-1)           # S = A correct,   B correct
18 T = (y_A * (1 - y_B)).sum(-1)     # T = A correct,   B incorrect
19 U = ((1 - y_A) * y_B).sum(-1)     # U = A incorrect, B correct
20 V = ((1 - y_A) * (1 - y_B)).sum(-1) # V = A incorrect, B incorrect
21
22 # calculate the bivariate normal mean
23 mu_As = norm(0,1).ppf(theta_As)
24 mu_Bs = norm(0,1).ppf(theta_Bs)
25
26 # Calculate probabilities of each cell in the 2x2 table
27 theta_V = binorm_cdf(x1=0, x2=0, mu1=mu_As, mu2=mu_Bs, sigma1=1, sigma2=1, rho=rhos)
28 theta_S = theta_As + theta_Bs + theta_V - 1
29 theta_T = 1 - theta_Bs - theta_V
30 theta_U = 1 - theta_As - theta_V
31
32 # Due to numerical issues, we need to handle the case where the probabilities are not in [0,1]
33 # (probabilities may be very small and negative instead of 0)
34 valid_idx = (theta_S > 0) & (theta_T > 0) & (theta_U > 0) & (theta_V > 0)
35 log_weights = S*np.log(theta_S[valid_idx]) + T*np.log(theta_T[valid_idx]) + \
36               U*np.log(theta_U[valid_idx]) + V*np.log(theta_V[valid_idx])
37
38 # normalise the weights
39 weights = np.zeros(K)
40 weights[valid_idx] = np.exp(log_weights - log_weights.max())
41 weights /= weights.sum()
42
43 # obtain samples from the posterior
44 posterior = (theta_As - theta_Bs)[np.random.choice(K, size=K, replace=True, p=weights)]
45
46 # Bayesian credible interval
47 bayes_ci = np.percentile(posterior, [2.5, 97.5])
```

# F. Computational Cost

In Fig. 11 we report the mean computation time of the different methods discussed in § 2.1, § 3.2, and § 3.4 averaged over 1000 repeated runs on a single CPU.

Whilst we observe the Bayesian (and bootstrap) methods taking longer than the CLT-based methods, in the few-data setting of this paper the computational cost of all of these methods is trivial. (The longest compute-time was 200 milliseconds for the largest Bayesian model in Fig. 3.) In the case where $N$ is much larger and compute cost starts to grow, the faster CLT-based methods would perform acceptably.

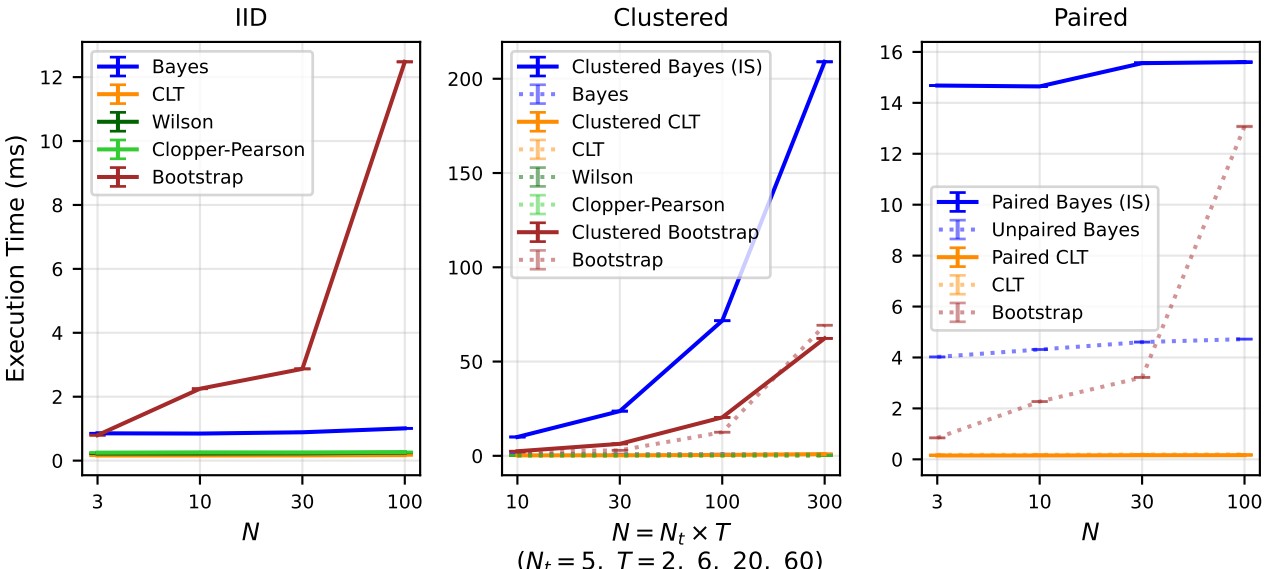

Figure 11: **Computational time.** Time (in milliseconds) required to compute error bars on a set of $N$ evals with different methods. Results are averaged over 1000 repeats on a single CPU, and error bars (which are close to zero) report standard errors.

## G. Robustness of Experiments to Random Seeding

In Fig. 12 we show an alternative version of Fig. 2 (comparing various methods in the IID setting of § 3.1) in which results are averaged over five runs using different random seeds. Error bars representing standard errors are shown in faint colours but are generally very small (on the order of $10^{-3}$ or smaller) and therefore often hard to identify. This suggests that the experiment methodology is robust to the randomness inherent in the data generating procedure.

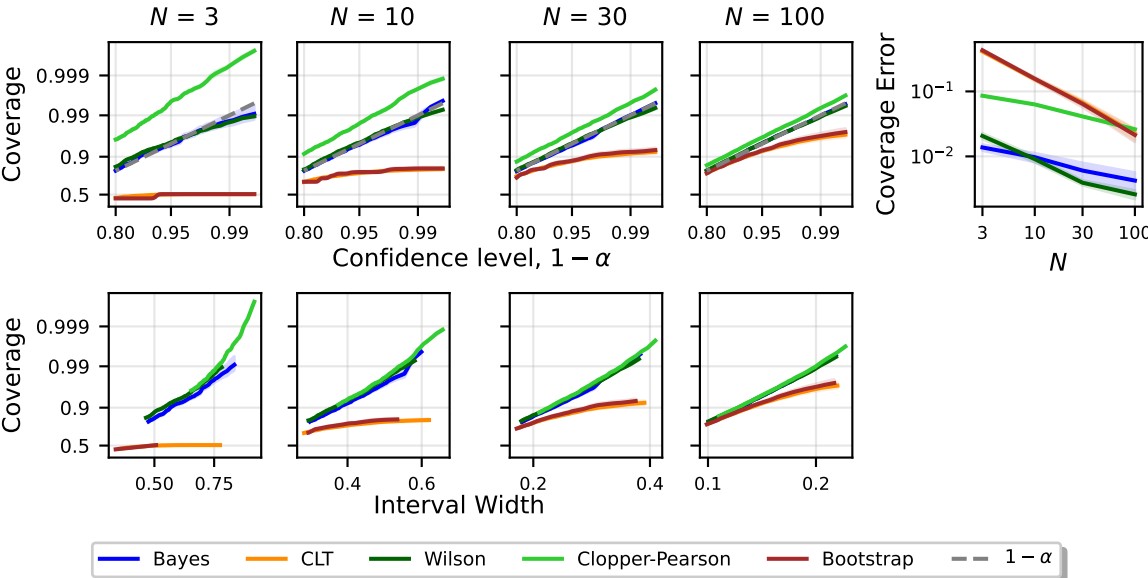

Figure 12: **IID question setting.** Coverage vs. confidence level (top) and coverage vs. width (bottom) for various methods. Coverage error denotes the mean absolute difference between true and nominal coverage for each method at a given value of $N$. **Error bars are given as faint lines around the means representing the standard error over 5 repeated experiments.** These error bars are very small, suggesting that each experiment successfully extracts values very close to true coverage per method.

## H. Interval Width vs. Coverage Plots

Here we present expanded versions of Fig. 3, Fig. 4 and Fig. 5 which include the original coverage vs. confidence level results in their top rows as well as a bottom row showing coverage vs. interval width. Fig. 13 presents these results for the clustered question setting of § 3.2, Fig. 14 shows the independent model comparison setting of § 3.3, and Fig. 15 shows the paired question setting of § 3.4.

In each case, we observe that, much like in § 3.1, the Bayesian methods tend to generate much more efficient (i.e. narrower) error bars for a given coverage than CLT- and bootstrap-based methods.

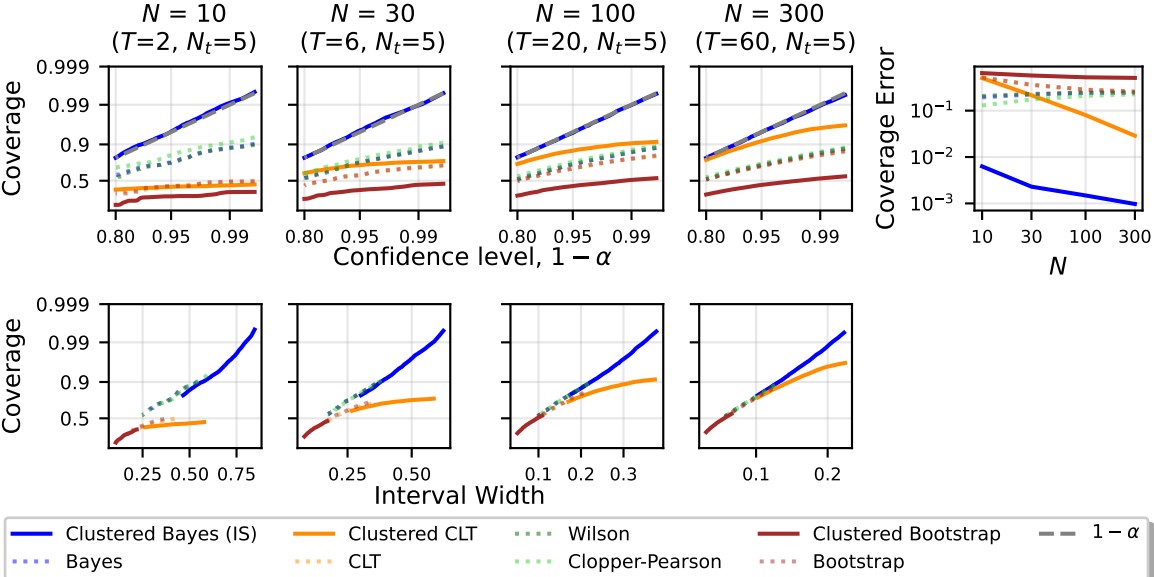

Figure 13: **Clustered questions setting**. Coverage vs. confidence level (**top**) and vs. interval-width (**bottom**) for various interval-calculation methods on the value of $\theta$. Methods ignoring the clustered structure of the data—assuming instead IID questions as per § 3.1—are shown as dotted lines. Results are averaged over 100 values of $\theta \sim \mathrm{Beta}(1, 1)$, each with 200 repeated experiments with randomly generated datasets.

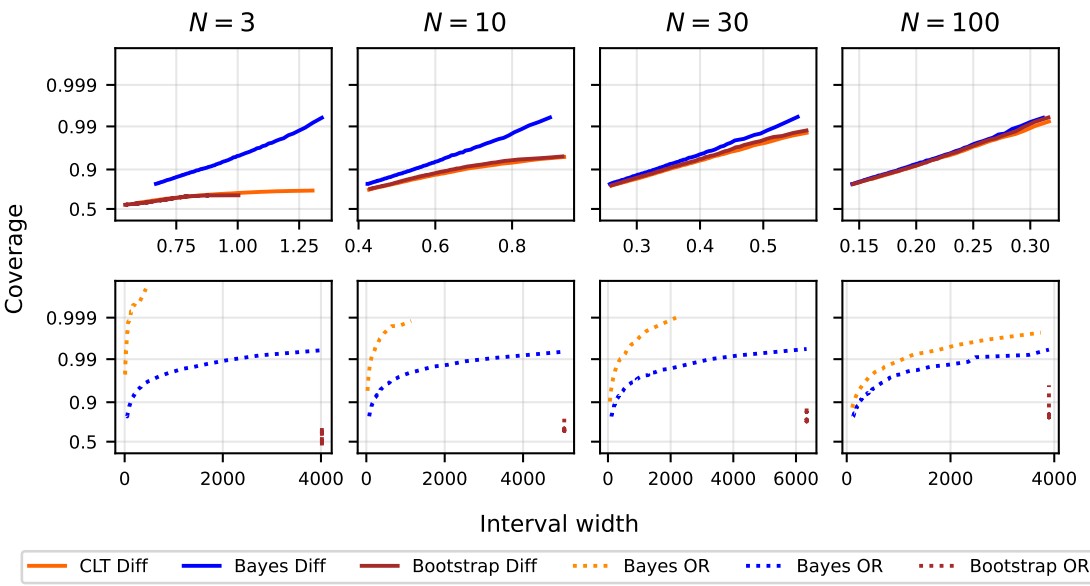

Figure 14: **Independent model comparison setting**. Coverage vs. interval-width for various interval-calculation methods. The top row includes intervals for the difference, $\theta_A - \theta_B$, whilst the bottom row shows intervals for the odds ratio. Note that in the odds ratio methods, we have dropped infinite width values for the method based on the Fisher exact test, which accounted for 43.5%, 17%, 6.5% and 1.9% for $N = 3, 10, 30$ and $100$, respectively. We have also clipped bootstrap widths as it produced extremely large (order $1e14$, essentially infinite) intervals.

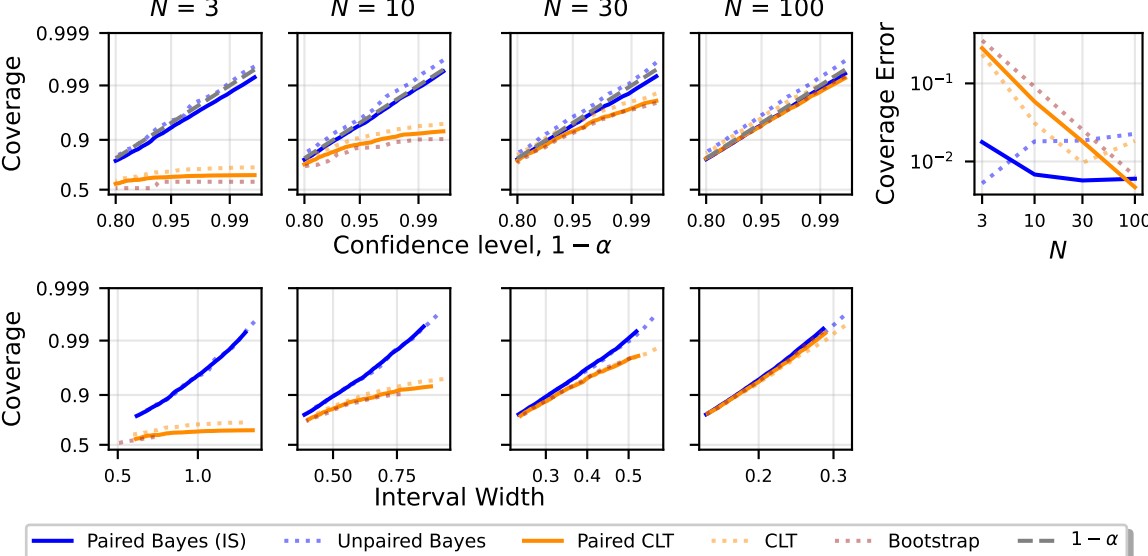

Figure 15: **Paired questions setting**. Coverage vs. confidence level (**top**) and vs. interval-width (**bottom**) for various interval-calculation methods on the value of $\theta_A - \theta_B$. Methods ignoring the paired structure of the data—assuming instead IID questions and answers from model A and from model B, as per § 3.1—are shown as dotted lines. Results are averaged over 100 values of $\theta_A, \theta_B \sim \mathrm{Beta}(1, 1)$, each with 200 repeated experiments with randomly generated datasets.

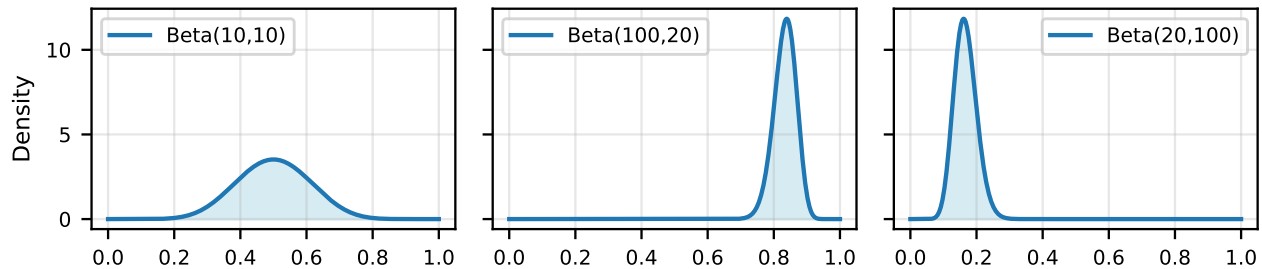

Figure 16: **Probability density functions**. Beta$(10, 10)$ **(left)**; Beta$(100, 20)$ **(centre)**; Beta$(20, 100)$ **(right)**.

## I. Ablations

### I.1. IID Questions Setting

We give the confidence-level vs. width and coverage plots for the setting presented in § 2.1 but where our data does not come from $\theta \sim \text{Beta}(1, 1) = \text{Uniform}(0,1)$. Specifically we consider three alternative true priors: Beta$(10, 10)$, $\theta \sim \text{Beta}(100, 20)$ and $\theta \sim \text{Beta}(20, 100)$, the probability density functions of which are shown in Fig. 16. These three settings are presented in Fig. 17, Fig. 18, and Fig. 19 respectively. As in § 2.1, results are averaged over 100 values of $\theta$ drawn the given prior, each used to generate 200 random datasets.

We can see that the mismatched prior does affect the coverage of each method, in particular leading to Bayesian credible intervals that are too wide. However, as the amount of data increases, this problem resolves fairly quickly.

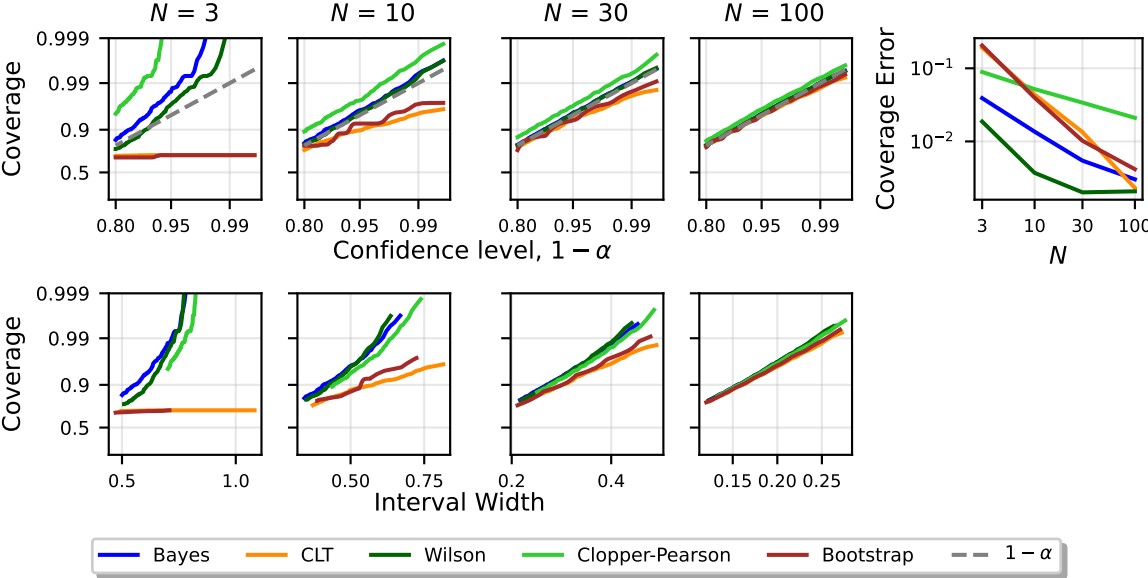

Figure 17: **IID question setting.** Coverage of intervals on $\theta$ with mismatched $\theta \sim \text{Beta}(10, 10)$ prior.

In Fig. 20, we also present plots for experiments with fixed $\theta$ values, specifically with $\theta \in \{0.5, 0.8, 0.95\}$. With each fixed $\theta$ value, we average results over 3000 simulated datasets. These show much the same behaviour as we saw in Figs. 17-19, but are useful in that they represent a more frequentist approach to the evaluations, as opposed to the typical Bayesian setting in which we treat the 'true' $\theta$ as random rather than fixed.

### I.2. Clustered Questions Setting

Similarly to the previous section, we show results in the clustered questions setting (§ 3.2) but with a mismatched true prior. Fig. 21 shows the results for $\theta \sim \text{Beta}(10, 10)$, Fig. 22 shows the results for $\theta \sim \text{Beta}(100, 20)$, and Fig. 23 shows the

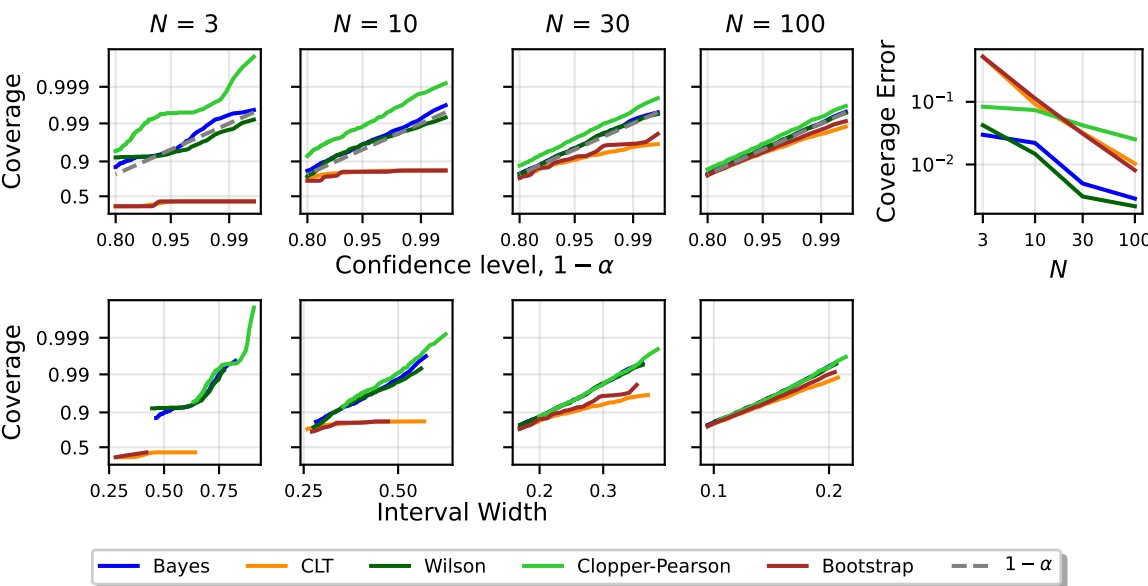

Figure 18: **IID question setting.** Coverage of intervals on $\theta$ with mismatched $\theta \sim \mathrm{Beta}(100, 20)$ prior.

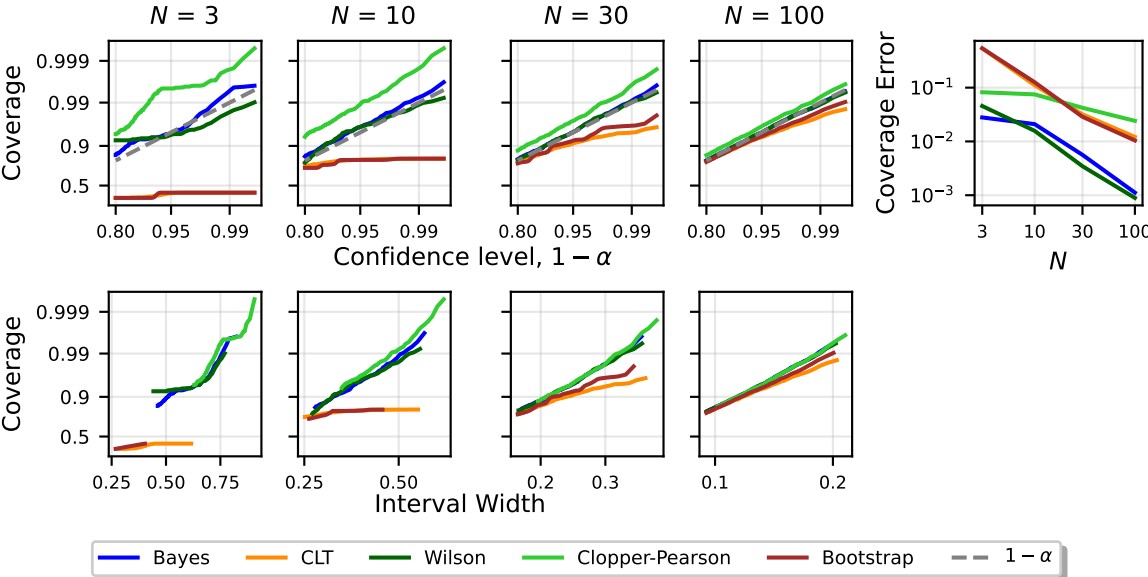

Figure 19: **IID question setting.** Coverage of intervals on $\theta$ with mismatched $\theta \sim \mathrm{Beta}(20, 100)$ prior.

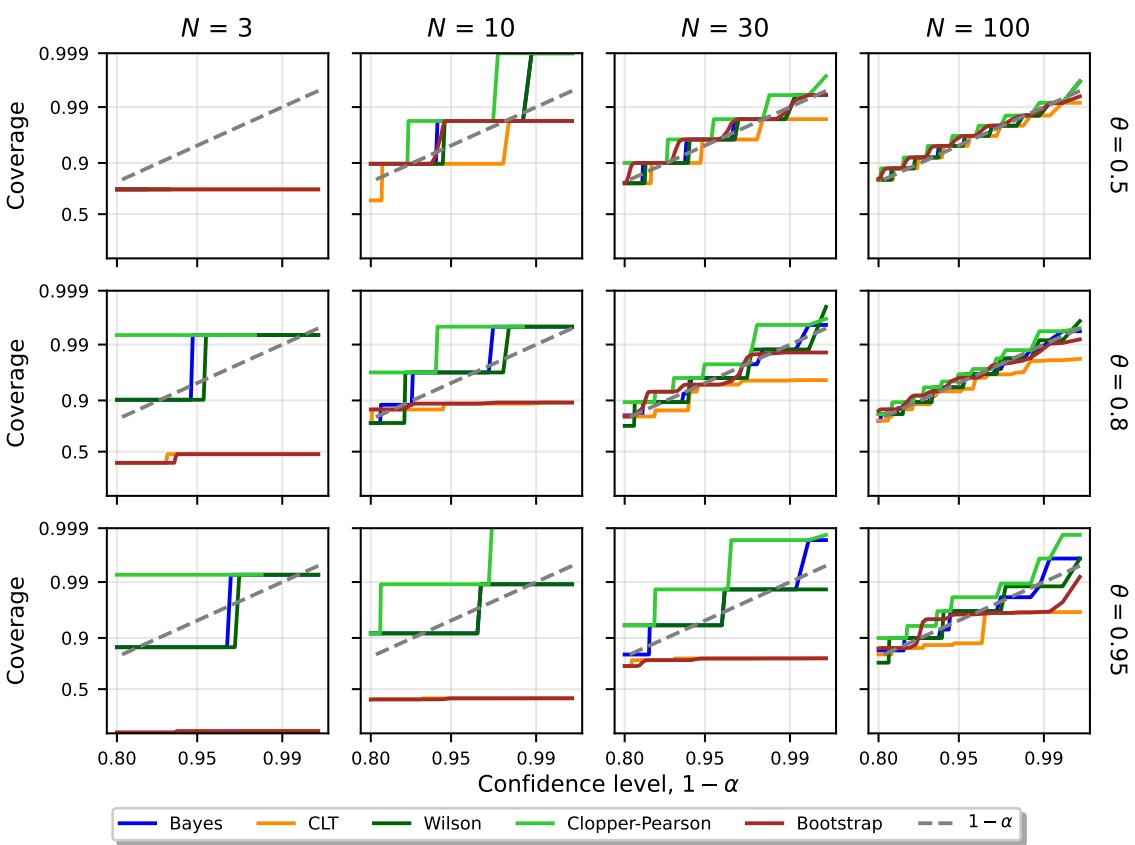

Figure 20: **IID question setting.** Coverage of intervals on $\theta$ generated with fixed values of $\theta \in \{0.5, 0.8, 0.95\}$.

results for $\theta \sim \text{Beta}(20, 100)$. As in § 3.2, results are averaged over 100 values of $\theta$ drawn the given prior, each used to generate 200 random datasets.

In each case, we observe close to ideal performance from the clustered Bayes credible intervals, whereas all other methods tend to struggle to match their nominal coverage for small $N$.

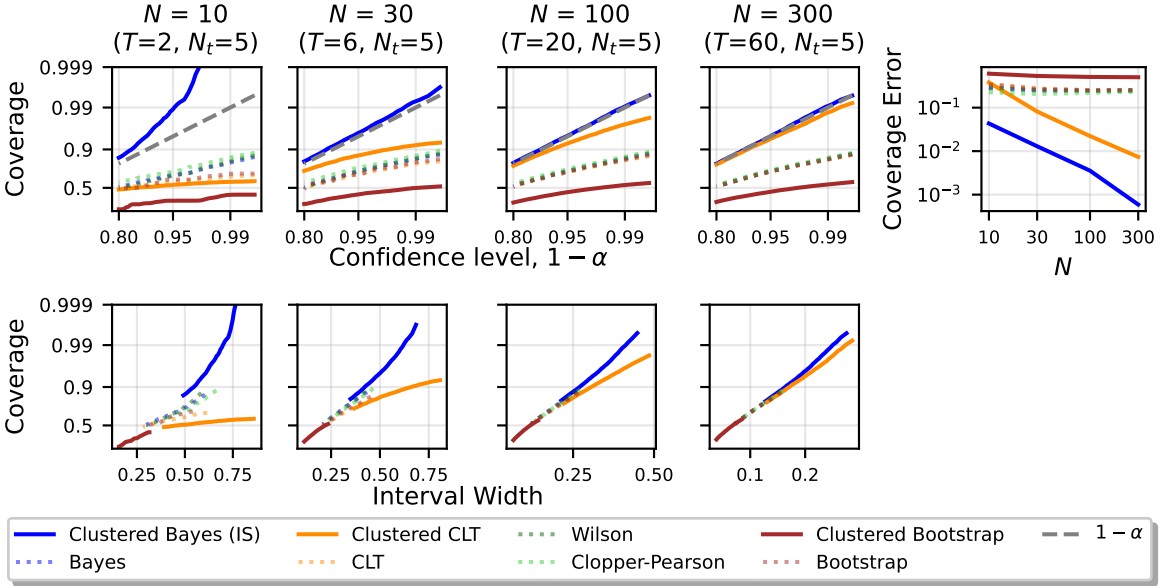

Figure 21: **Clustered question setting.** Coverage of intervals on $\theta$ with mismatched $\theta \sim \text{Beta}(10, 10)$ prior.

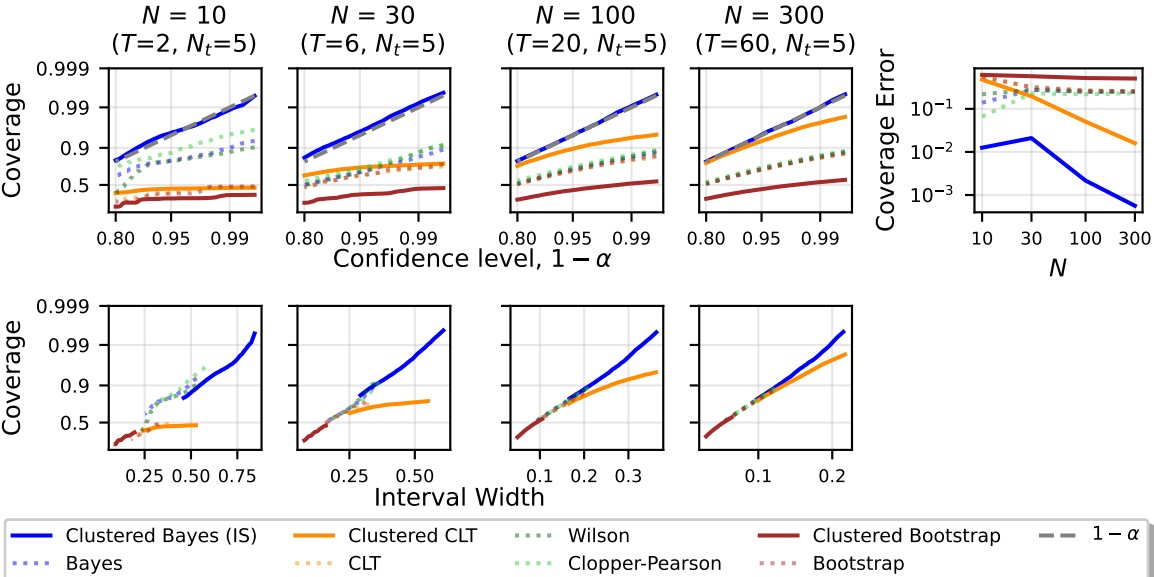

Figure 22: **Clustered question setting.** Coverage of intervals on $\theta$ with mismatched $\theta \sim \text{Beta}(100, 20)$ prior.

Again, we present results generated from fixed values of $\theta \in \{0.5, 0.8, 0.95\}$ rather than from a Bayesian prior. With each fixed $\theta$ value, we average results over 3000 simulated datasets (each with a different sampled dispersion parameter $d \sim \text{Gamma}(1, 1)$. These can be seen in Fig. 24 and show much the same behaviour as the previous plots.

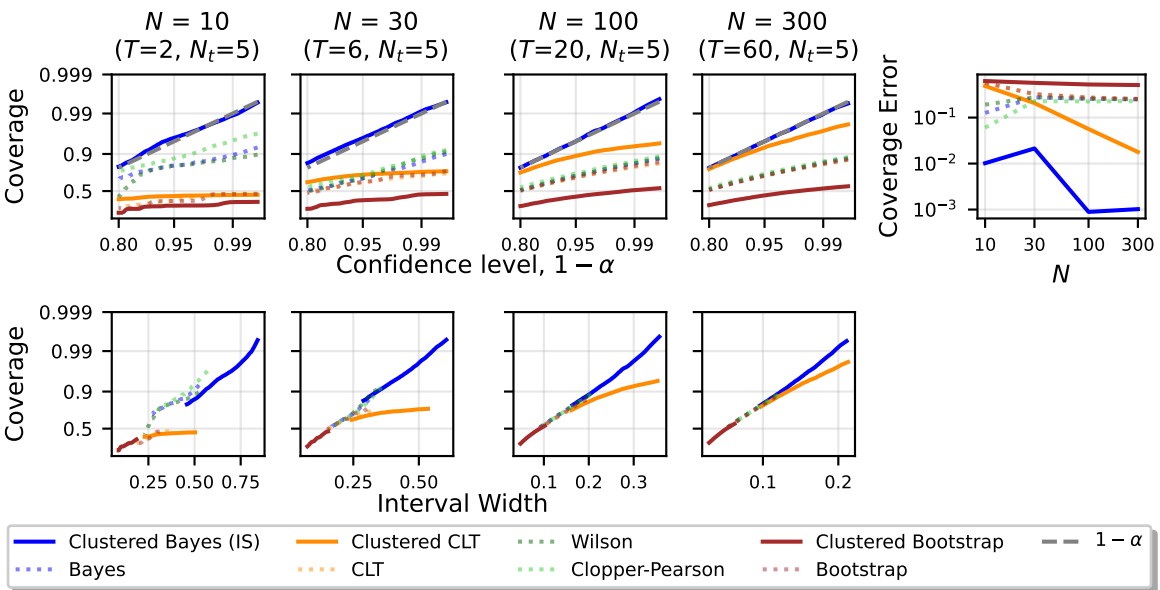

Figure 23: **Clustered question setting.** Coverage of intervals on $\theta$ with mismatched $\theta \sim \mathrm{Beta}(20, 100)$ prior.

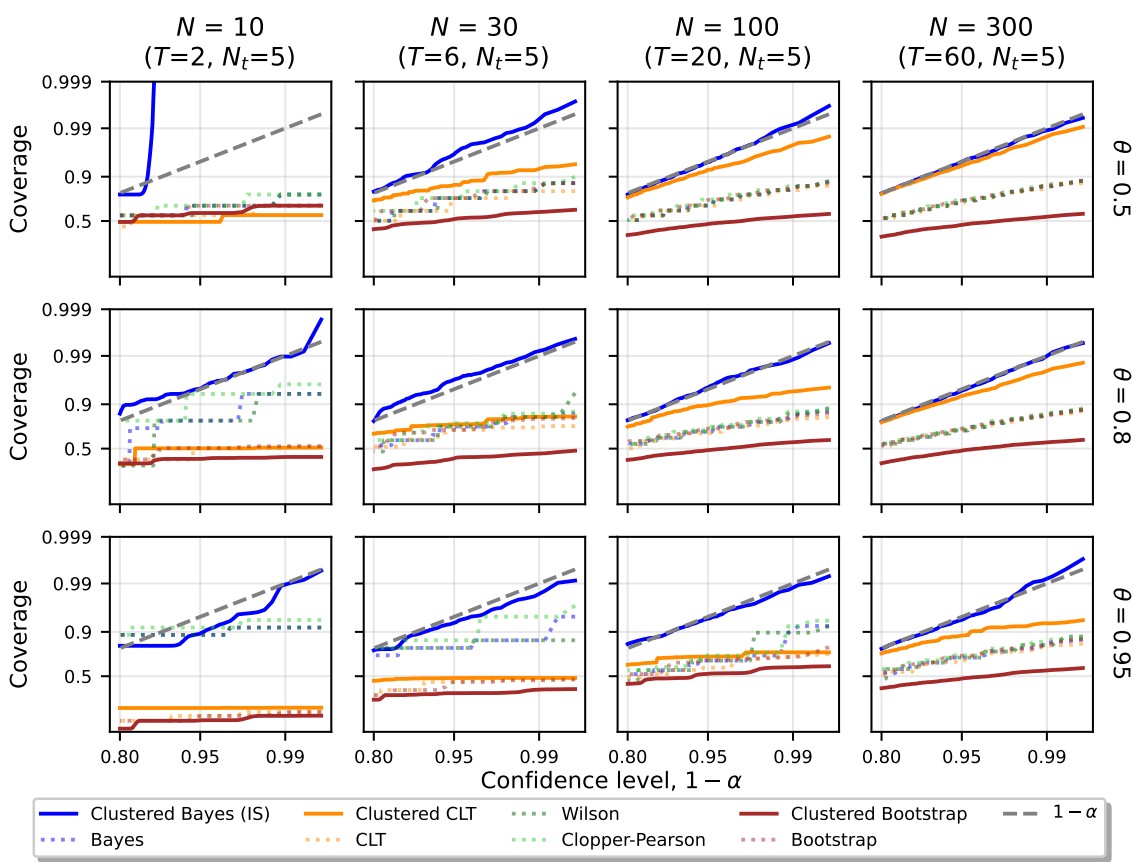

Figure 24: **Clustered question setting.** Coverage of intervals on $\theta$ generated with fixed values of $\theta \in \{0.5, 0.8, 0.95\}$.

## I.3. Independent Model Comparison

Similarly to § I.1 and § I.2, we show results for the independent model comparison setting (§ 3.3) but with a mismatched true prior, i.e. the actual data generating process does not match the prior that we assume for our the Bayesian model.

Specifically, we consider the following settings:

- Neither model A nor model B has a uniform prior: Fig. 25a shows results in which both $\theta_A, \theta_B \sim \text{Beta}(100, 20)$, whilst Fig. 25b presents results in which we have $\theta_A \sim \text{Beta}(100, 20)$ and $\theta_B \sim \text{Beta}(20, 100)$

- Keep the same prior for $\theta_A$, that is, $\text{Beta}(1, 1) = \text{Uniform}[0, 1]$ and vary the prior for $\theta_B$ between $\text{Beta}(100, 20)$ (Fig. 26a); $\text{Beta}(10, 10)$ (Fig. 26b); and $\text{Beta}(20, 100)$ (Fig. 26c).

- Fixed values for $\theta_A$ and $\theta_B$ as follows: $\theta_A \in \{0.5, 0.8, 0.9\}$ with $\theta_B$ taking values in $\{\theta_A, \theta_A - 0.3, \theta_A - 0.8\}$ in each setting. These are shown in Fig. 27, Fig. 28, and Fig. 29 respectively. Results are shown averaged over 3000 simulated datasets for each $(\theta_A, \theta_B)$ pair.

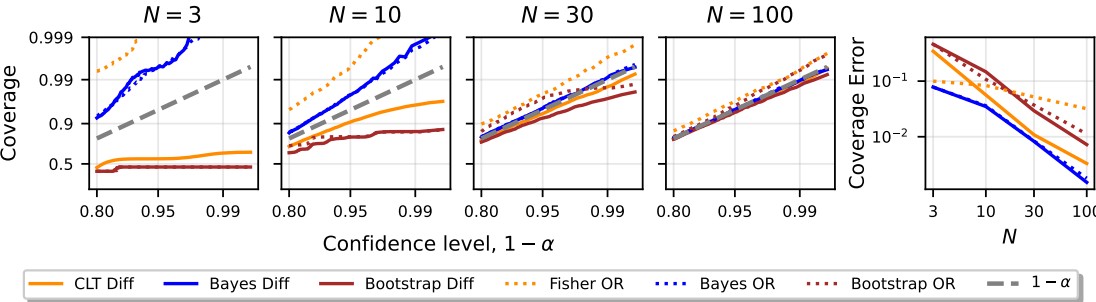

(a) Prior used to generate data: $\theta_A, \theta_B \sim \text{Beta}(100, 20)$

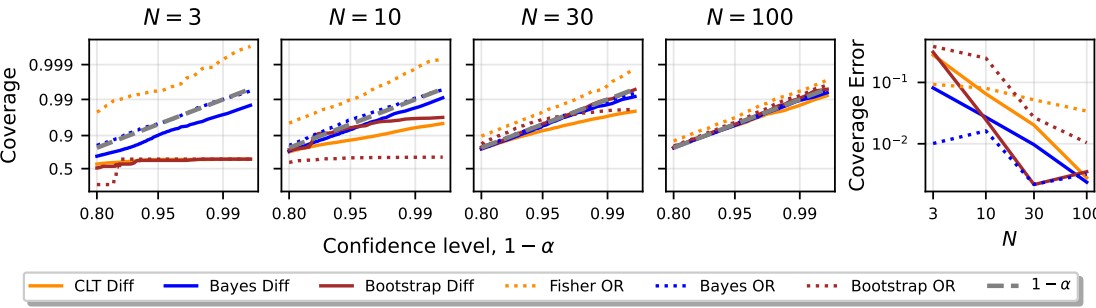

(b) Prior used to generate data: $\theta_A \sim \text{Beta}(100, 20)$ and $\theta_B \sim \text{Beta}(20, 100)$.

Figure 25: **Independent model comparison.** Prior mismatch: we use uniform, $\text{Beta}(1, 1)$, prior in the Bayesian model used to construct confidence intervals, whilst at test time the data is generated using a different prior.

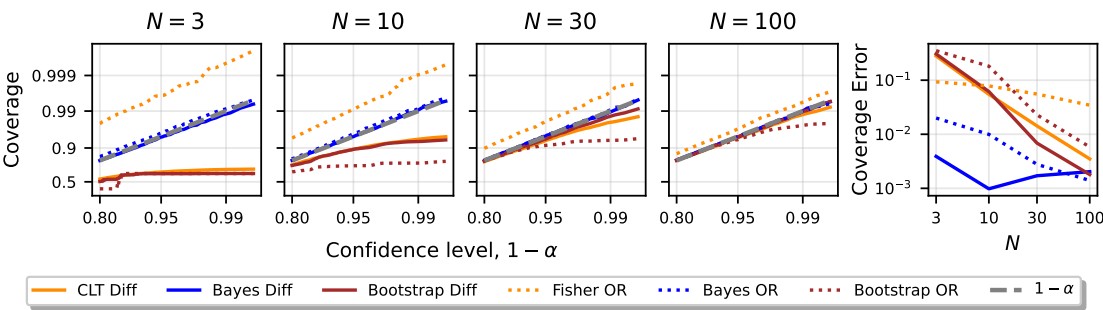

(a) Prior used to generate data: $\theta_B \sim \text{Beta}(100, 20)$

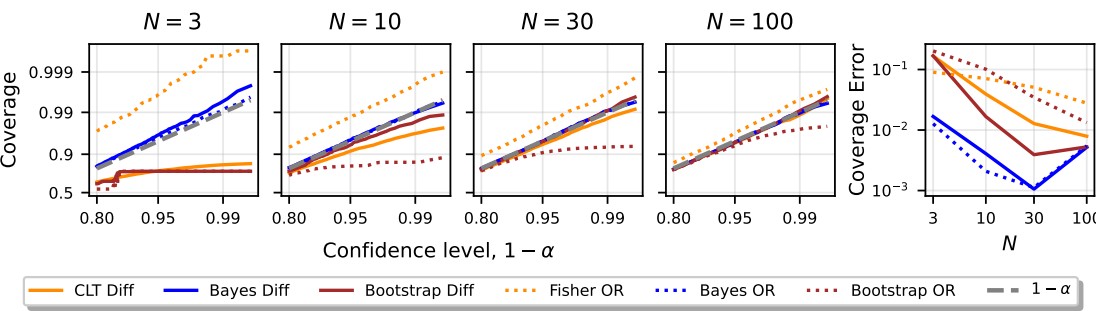

(b) Prior used to generate data: $\theta_B \sim \text{Beta}(10, 10)$

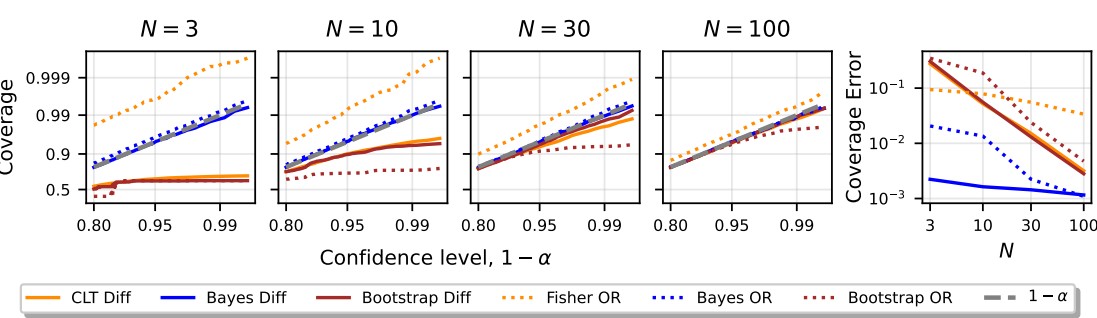

(c) Prior used to generate data: $\theta_B \sim \text{Beta}(20, 100)$

Figure 26: **Independent model comparison.** Prior mismatch: we use uniform, $\text{Beta}(1, 1)$, prior in the Bayesian model used to construct confidence intervals. At test time, the accuracy of model A is sampled from $\text{Beta}(1, 1)$ whilst that of model B is sampled from a different prior.

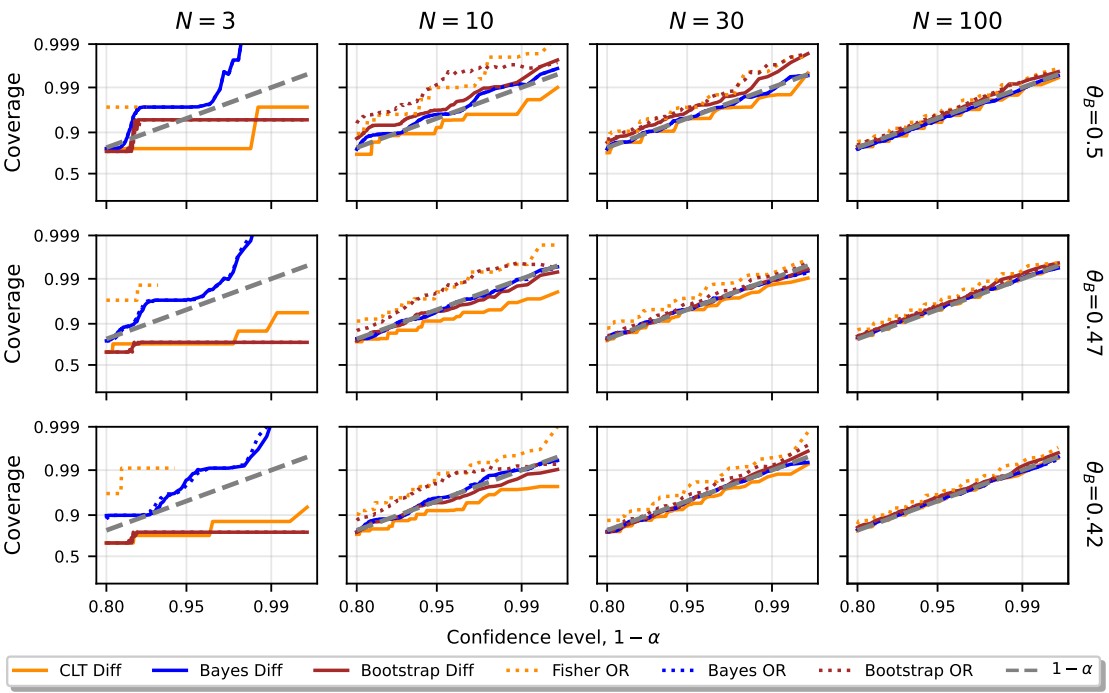

Figure 27: **Independent model comparison.** Success probabilities are fixed at $\theta_A = 0.5$ and $\theta_B \in \{0.5, 0.47, 0.42\}$.

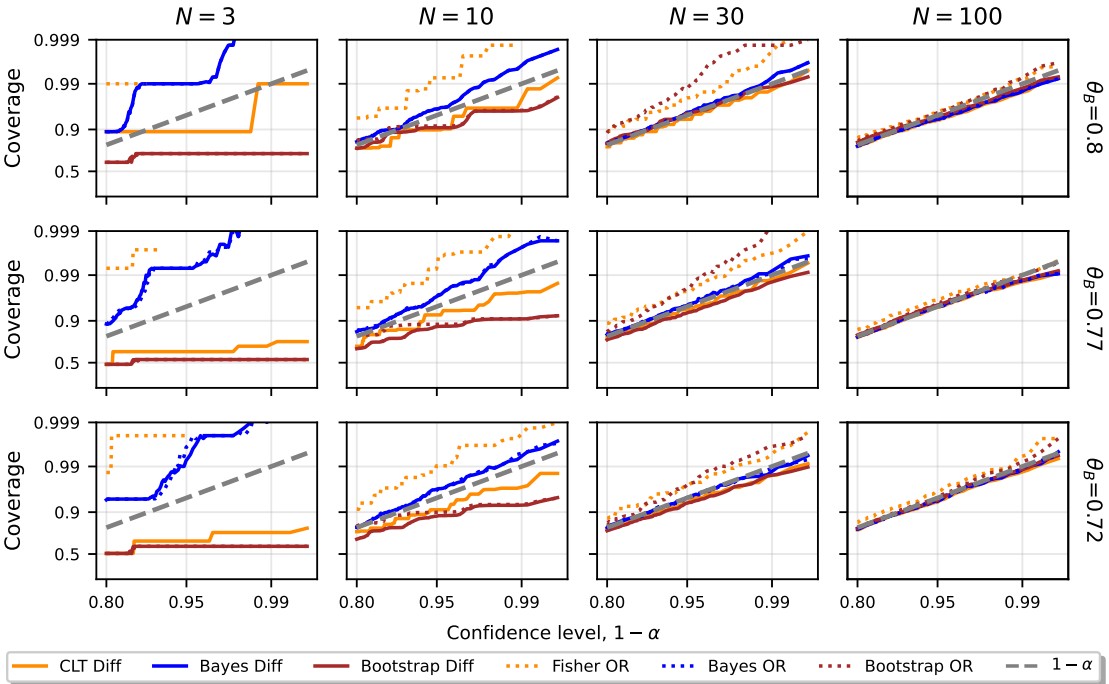

Figure 28: **Independent model comparison.** Success probabilities are fixed at $\theta_A = 0.8$ and $\theta_B \in \{0.8, 0.77, 0.72\}$.

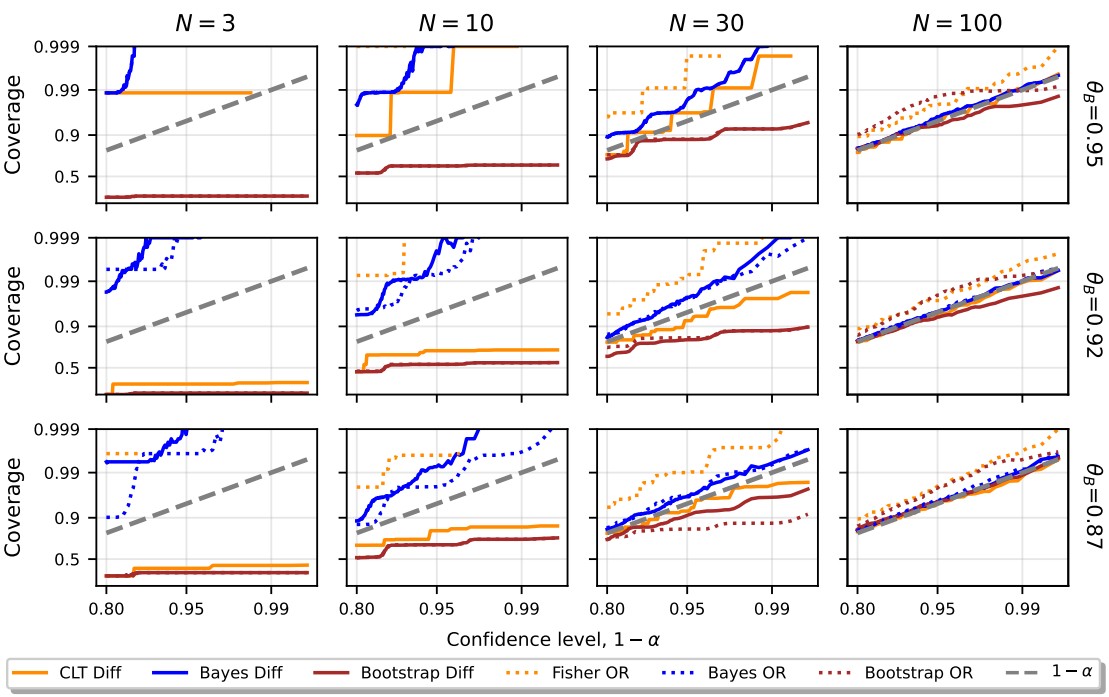

Figure 29: **Independent model comparison.** Success probabilities are fixed at $\theta_A = 0.95$ and $\theta_B \in \{0.95, 0.92, 0.87\}$.

### I.4. Paired Questions Setting

Similarly to § I.1 and § I.2, we show results for the paired questions setting (§ 3.4) but with a mismatched true prior. In particular, we keep the same prior for $\theta_A$ (that is, $\text{Beta}(1, 1) = \text{Uniform}[0, 1]$) but vary the prior for $\theta_B$ between $\text{Beta}(10, 10)$ (Fig. 30); $\text{Beta}(100, 20)$ (Fig. 31); and $\text{Beta}(20, 100)$ (Fig. 32). We also present results where neither model A nor model B has a uniform prior: Fig. 33 shows results in which both $\theta_A, \theta_B \sim \text{Beta}(100, 20)$, whilst Fig. 34 presents results in which we have $\theta_A \sim \text{Beta}(100, 20)$ and $\theta_B \sim \text{Beta}(20, 100)$. As in § 2.1, results are averaged over 100 values of $(\theta_A, \theta_B)$ drawn the specified priors, with each $(\theta_A, \theta_B)$ pair used to generate 200 random datasets.

In each case, we observe the coverage of Bayesian credible intervals approaching the ideal $1 - \alpha$ line at least as quickly (in terms of an increasing $N$) as the paired-CLT confidence intervals. We also observe the paired Bayesian credible intervals generally outperforming the unpaired Bayesian intervals.

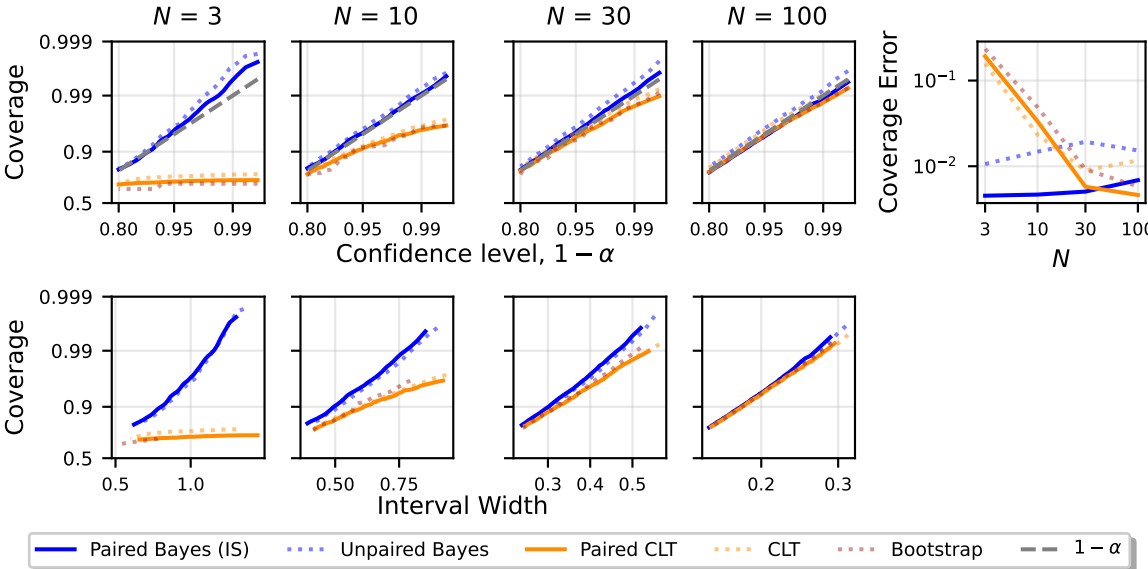

Figure 30: **Paired question model comparison setting.** Coverage of intervals on $\theta_A - \theta_B$ with mismatched $\theta_B \sim \text{Beta}(10, 10)$ prior and $\theta_A \sim \text{Beta}(1, 1)$.

Finally, for this section, we also present results with the same fixed values for $\theta_A$ and $\theta_B$ as in § I.3: $\theta_A \in \{0.5, 0.8, 0.9\}$ with $\theta_B$ taking values in $\{\theta_A, \theta_A - 0.3, \theta_A - 0.8\}$ in each setting. Results are shown averaged over 3000 simulated datasets for each $(\theta_A, \theta_B)$ pair. These are shown in Fig. 35, Fig. 36, and Fig. 37 respectively.

Whilst the coverage of both the Bayesian credible intervals and the paired CLT-based confidence intervals are very similar when $\theta_A = \theta_B$, we see much more robust behaviour from the Bayesian method when we increase the difference between $\theta_A$ and $\theta_B$.

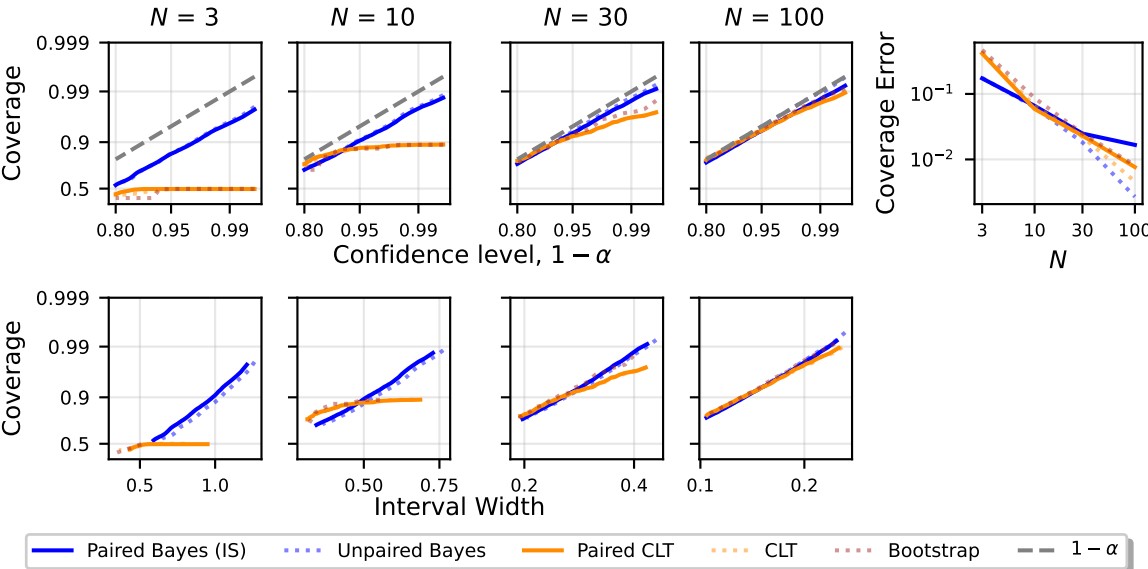

Figure 31: **Paired question model comparison setting.** Coverage of intervals on $\theta_A - \theta_B$ with mismatched $\theta_B \sim$ Beta$(100, 20)$ prior and $\theta_A \sim$ Beta$(1, 1))$

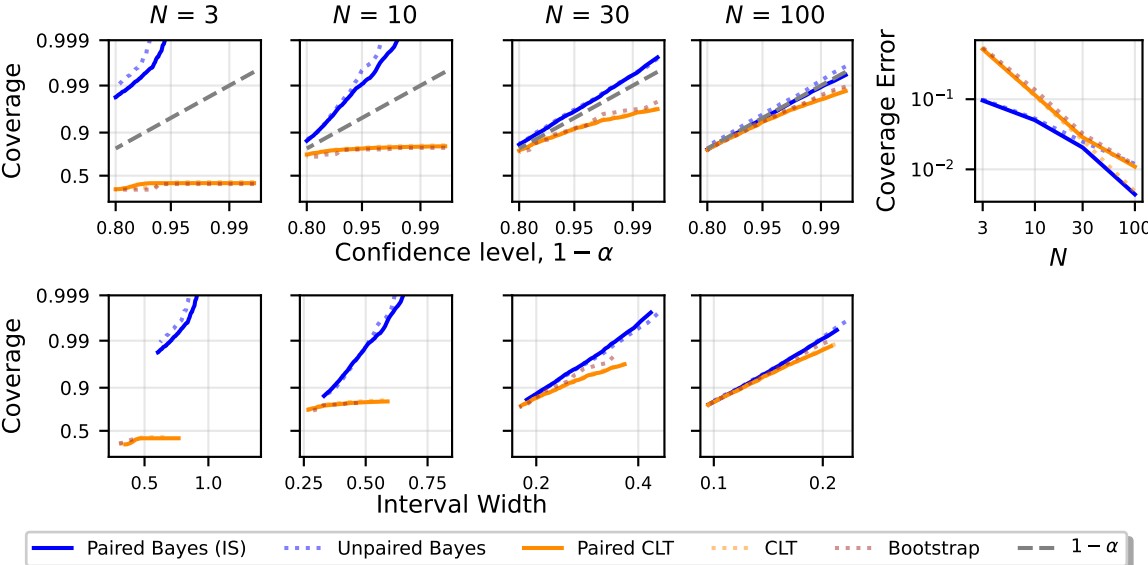

Figure 32: **Paired question model comparison setting.** Coverage of intervals on $\theta_A - \theta_B$ with mismatched $\theta_B \sim$ Beta$(20, 100)$ prior and $\theta_A \sim$ Beta$(1, 1)$.

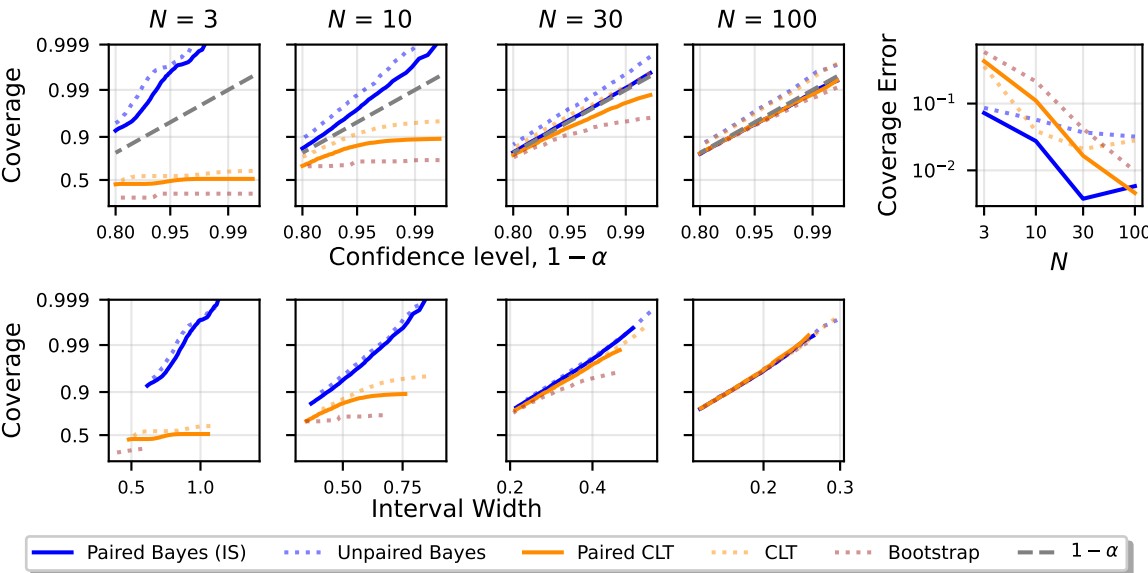

Figure 33: **Paired question model comparison setting.** Coverage of intervals on $\theta_A - \theta_B$ with mismatched $\theta_A, \theta_B \sim \text{Beta}(20, 100)$ priors.

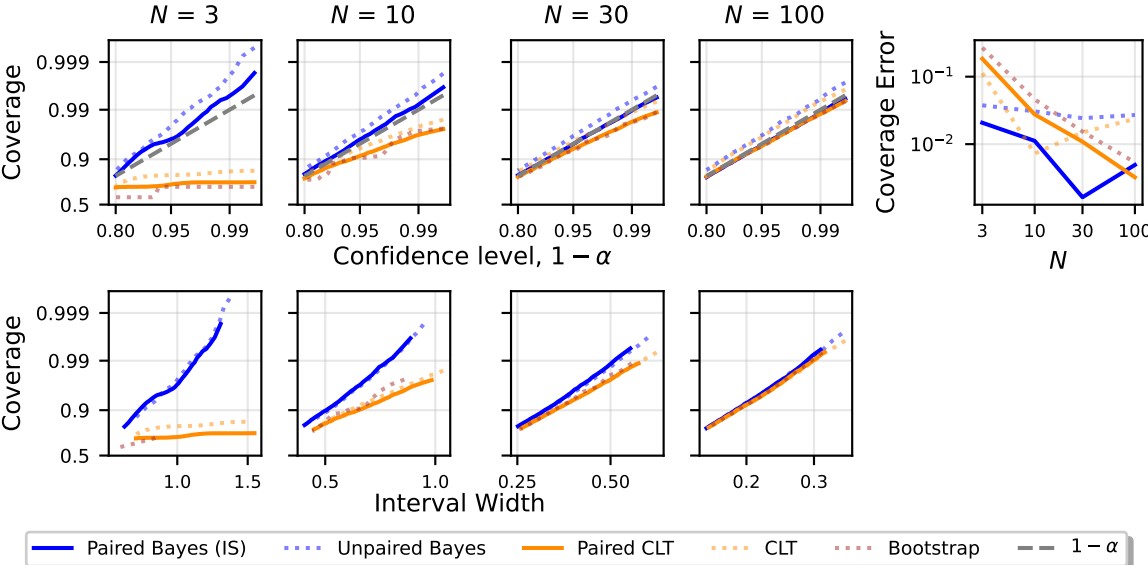

Figure 34: **Paired question model comparison setting.** Coverage of intervals on $\theta_A - \theta_B$ with mismatched priors: $\theta_A \sim \text{Beta}(100, 20)$, and $\theta_B \sim \text{Beta}(20, 100)$.

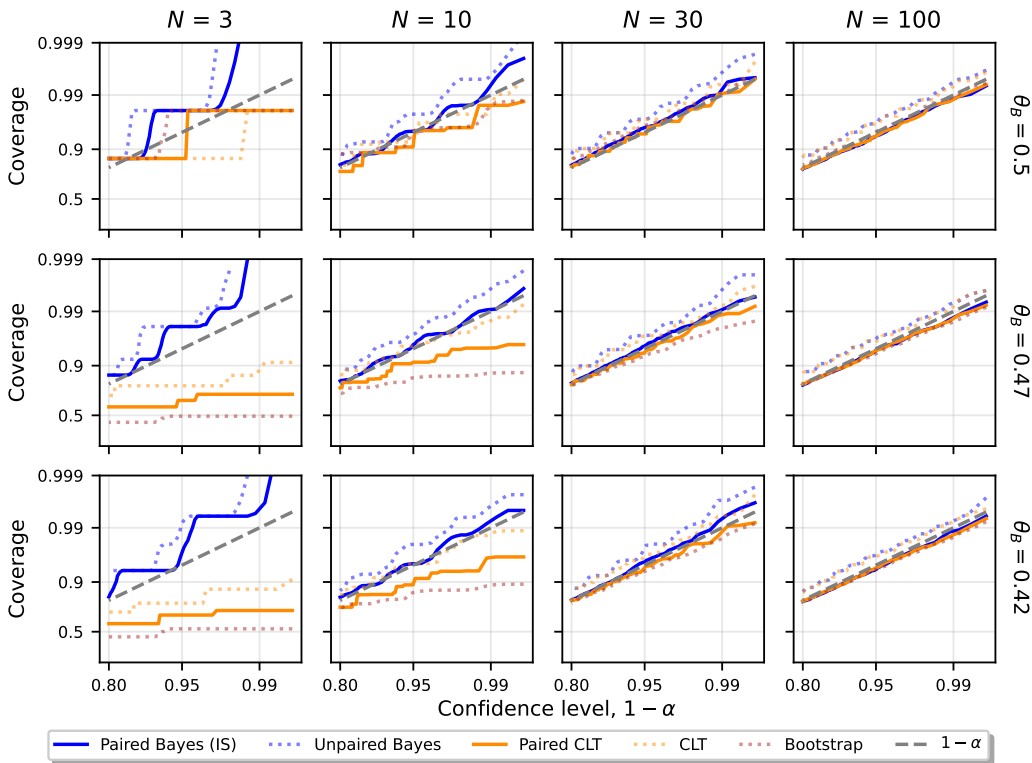

Figure 35: **Paired question model comparison setting.** Coverage of intervals on $\theta_A - \theta_B$ generated with $\theta_A = 0.5$ and $\theta_B \in \{0.5, 0.47, 0.42\}$.

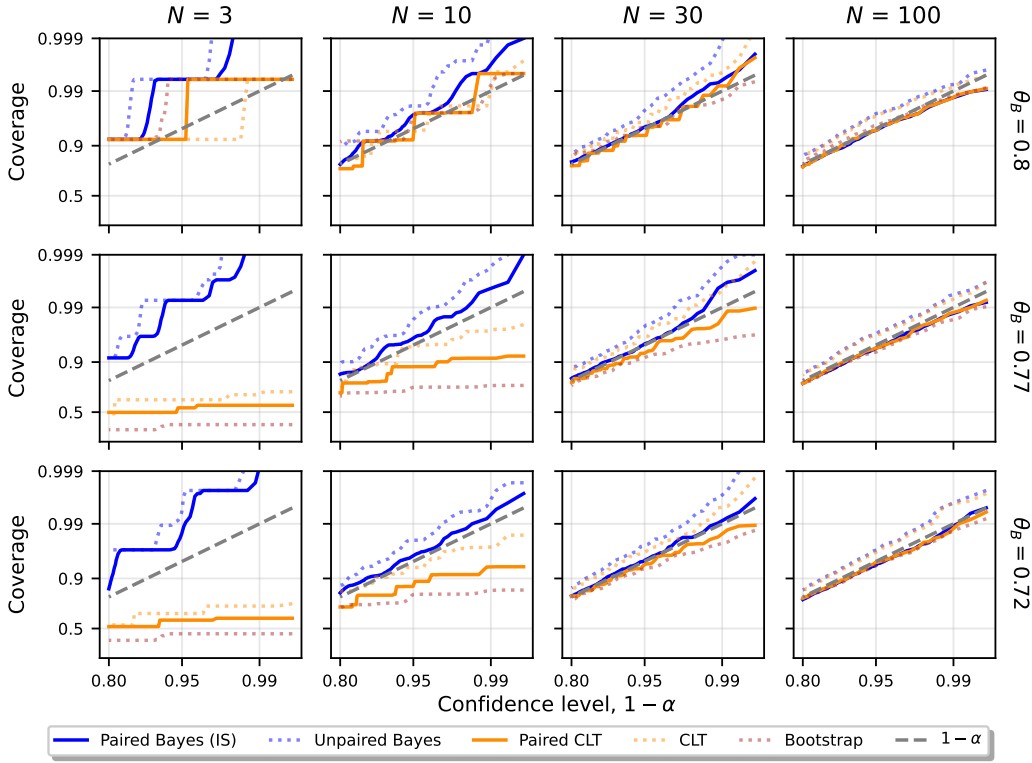

Figure 36: **Paired question model comparison setting.** Coverage of intervals on $\theta_A - \theta_B$ with $\theta_A = 0.8$ and $\theta_B \in \{0.8, 0.77, 0.72\}$.

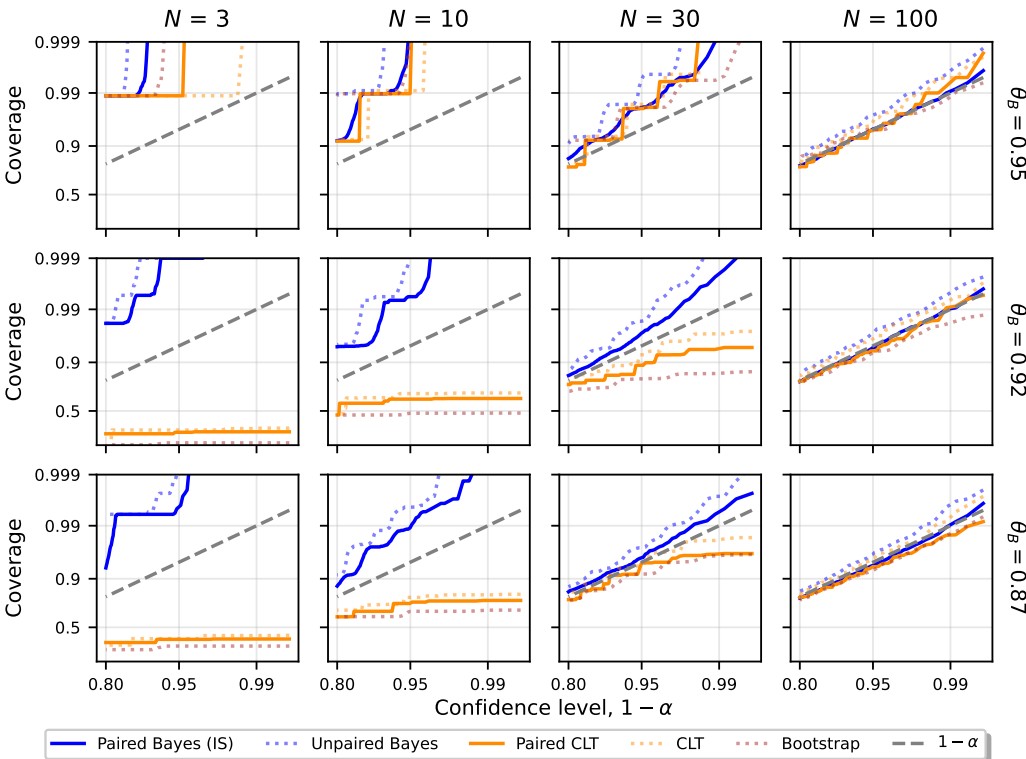

Figure 37: **Paired question model comparison setting.** Coverage of intervals on $\theta_A - \theta_B$ with $\theta_A = 0.95$ and $\theta_B \in \{0.95, 0.92, 0.87\}$.

## J. Error Bars on the $F_1$ Score

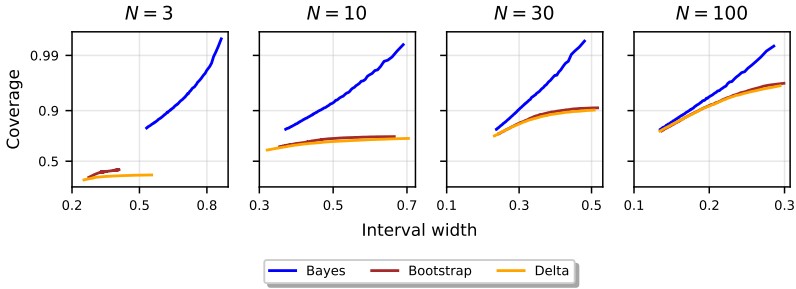

Figure 38: **Error bars for the $F_1$-score.** Coverage vs. interval-width for various interval-calculation methods.

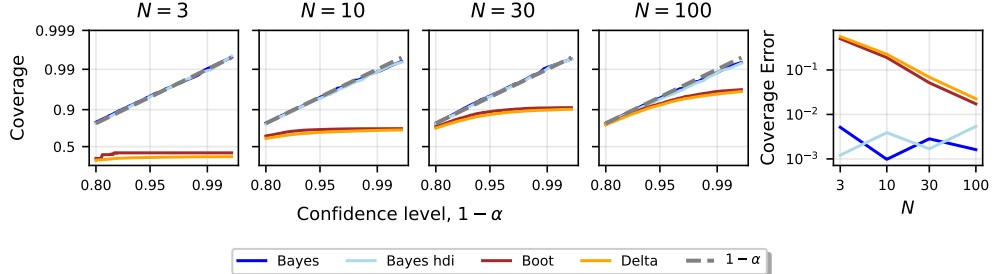

Figure 39: **Error bars for the $F_1$-score.** Coverage vs. confidence level for Bayesian and bootstrap intervals on $F_1$ scores using highest posterior density interval (HDI) instead of quantile-based interval (QBI), which was presented in Fig. 6.

