# OpenReview forum: "Position: Don't Use the CLT in LLM Evals With Fewer Than a Few Hundred Datapoints"
_ICML.cc/2025/Position_Paper_Track — ICML 2025 Position Paper Track spotlightposter_

### Official Review · Reviewer_TPwH · 2025-02-16

**Significance:** 4
**Argument Clarity:** 3
**Rating:** 4
**Confidence:** 4

**Questions:**

No further questions.

**Discussion Potential:**

4

**Paper Summary:**

The paper argues that central limit theorem (CLT)-based confidence intervals should not be used in LLM evaluations that have a small sample size. Instead, the paper advocates using frequentist or Bayesian interval estimates that do not need a large sample. The paper supports this position with simulations that evaluate the coverage of confidence (or credible) intervals. The simulations cover several practically relevant LLM evaluation setups, such as evaluating the accuracy of a single model, or comparing two models. In all simulations, CLT-based confidence intervals fall significantly short of the advertised coverage when the sample size is very small.

## update after rebuttal

The rebuttal addressed my concerns, as detailed in my response, so I've raised my score.

**Position:**

Yes

**Position In Title:**

Yes

**Related Work:**

4

**Strengths And Weaknesses:**

# Strengths
The topic of the paper is extremely relevant and timely due to the growing number of LLMs that need to be compared with each other. The writing of the paper is good, and the arguments are made clearly. The paper considers several practically relevant evaluation setups in the simulations. The results clearly support the position for the very small sample sizes.

# Weaknesses
The most glaring issue in the paper is the lack of error bars (or bands) on all of the results. This undermines the paper's assumption that LLM evaluations should have error bars. Afterall, if that is the case, why would evaluations of evaluations not need error bars?

A second weakness is that the sample sizes where the CLT fails are lower than many of the real evaluation datasets mentioned in the paper. For example, in the clustered evaluation setting, the CLT performs fairly well with $N = 300$ samples, where there are 60 clusters with 5 samples per cluster. This seems to be the closest setting to 4 out of 6 example datasets mentioned in the Introduction. In addition, the CLT performs well with $N = 100$ in all other settings, so the title's claim that fewer than a few hundred samples is too little for the CLT seems to be an exaggeration.

I think these weaknesses are barely significant enough to not accept the paper as is, but I'm very open to changing my recommendation if these are addressed in the rebuttal.

Smaller points:
- A table summarising the results would be useful. For example, you could have a check mark for methods that worked well per setting, and possibly a sample size threshold showing when CLT is safe to use.
- Figure 2, bottom row is very hard to read since it has overlapping lines that do not seem to cover the whole x-axis in some cases.
- $\theta$ subscripts are inconsistent in Appendix A.2 ("-" and "lower"; "+", "upper" and "higher").
- Figures 22, 23 and 24 seem to be missing $N  = \mathrm{number}$ titles.

**Support:**

3

---

> ### Author Rebuttal · Authors · 2025-03-31
>
> Thanks for your careful and considered review!
>
> We have added error bars to all the plots in our working draft.  The error bars are so small (their magnitude is on the order of $10^{-3}$ or smaller) that you can't actually see them (they are represented as shaded areas).  But they are there. Thus, these error bars do not change any of our results ([Rebuttal Figs 2,3,4,5](https://anonymous.4open.science/api/repo/CLT_rebuttal-CC75/file/rebuttals.pdf?v=c3156a4e)).
>
> ### Number of samples required for the CLT
> We agree with your points!  However, it is worth being aware of two considerations.
>
> First, our paper is a position paper, aimed at changing widespread practices in the community among LLM experts who aren't also experts in statistics.  Poor statistical practices are widespread in this community, including not giving error bars at all on evals with a tiny number of samples. Our title therefore had several goals:
>
> * to be very clear
> * to explicitly mention the CLT
> * to give a ballpark range for the number of samples where you have to worry
>
> So we could alternatively use the title "Be really careful using the CLT in LLM evals with fewer than a few hundred datapoints" (and we could actually use this title: let us know if you'd like us to make this change).  But we feel this title is less clear, and the "be really careful" isn't actually helpful in practice.  Specifically, "being really careful" is likely to involve evaluating the quality of your CLT-based error bars relative to e.g. Bayes or WS intervals.  And this evaluation (that we've done here) is _alot_ of work.  Far more work than just computing the Bayes or WS intervals.
>
> Moreover, you might argue that we could just reduce the number in the title.  For instance one could suggest the title "Don't use the CLT in LLM evals with fewer than (30/100) datapoints".  However, we worry that this will get translated to "It's fine to use the CLT in evals with more than (30/100) datapoints".  And we aren't comfortable with this claim, especially given that it may be applied in a variety of complex settings where more datapoints are needed.
>
> Second, differences in coverage that appear small in these plots can actually be highly relevant in practice. Perhaps the clearest evaluation is in Figure 2, where a confidence level of 0.95 for N=100 corresponds to a CLT-based coverage of 0.925 (i.e. the CLT error bars undercover).  This is a pretty big difference!  For reference, a two-tailed Gaussian interval with confidence 0.95 is “mean +/- 1.96\*stddev”, whereas confidence 0.925 would be “mean +/- 1.78\*stddev”. This difference is pretty big!
>
> > A table summarising the results would be useful. For example, you could have a check mark for methods that worked well per setting, and possibly a sample size threshold showing when CLT is safe to use.
>
> Agreed. We have added [Rebuttal Table 1](https://anonymous.4open.science/api/repo/CLT_rebuttal-CC75/file/rebuttals.pdf?v=c3156a4e) .
>
> > Figure 2, bottom row is very hard to read since it has overlapping lines that do not seem to cover the whole x-axis in some cases.
>
> Thanks!  It is always a little bit difficult to see what's going on when lines overlap so closely, so we have added a note about which lines are overlapping the caption. The difference in range on the x-axis arises because we do not specify the interval range in our tests.  Instead, we specify a range of confidence levels from 0.8 to 1.0 (in the x-axis of the top row), and that implies a range of interval widths.  Thus, the differences in the starting point for the interval-width arise because of differences in the mapping between from confidence-level to interval widths.  This difference is evident from the top-row, which displays confidence-level against coverage (which is closely related to interval width, as wider intervals have higher coverage).
>
> > subscripts are inconsistent in Appendix A.2 ("-" and "lower"; "+", "upper" and "higher").
>
> Thanks! Fixed.
>
> >Figures 22, 23 and 24 seem to be missing  titles.
>
> Thanks! Fixed.
>
> Thank you for generously outlining the paper's strengths in your original review. We hope this response (especially the new plots with error bars and the in-depth discussion of the number-of-samples) has addressed your key concerns. If so, we would greatly appreciate it if you would reconsider your score. As mentioned above, we would be happy to make any specific change to the title / text you would suggest around the number-of-samples point.

---

> > ### Comment · Reviewer_TPwH · 2025-04-02
> >
> > Thank you for the response. Your addition of the error bars is very appreciated, as are the more minor changes you have promised. I also broadly agree with your reasoning about the title, which is why I'm raising my score.
> >
> > There is still something I would like to say about the title. I agree that "Don't use" is better than something like "Be very careful" for the reasons you have given. I also agree that "don't use with fewer than 100 datapoints" is easily translated to "fine with more than 100 datapoints", so changing the number is also not a good option.
> >
> > However, the downside is that since you do not clearly show failure cases with a few hundred datapoints, it is easy to claim that the paper is exaggerating the problem. Now, if an LLM expert hears about the paper, but also hears that it is exaggerating, they could easily dismiss the paper, even if they were using evaluations with much fewer than a hundred datapoints. To get ahead of this, I think you should include your reasoning for the title from your rebuttal in the paper.
> >
> > Also, I recognize that I underestimated the undercoverage of the CLT in Figure 2, and I would expect many others to do the same. I wonder if you could highlight this difference in the paper in some way, for example by somehow plotting the interval width difference you gave in the rebuttal.

---

> > > ### Author Response · Authors · 2025-04-02
> > >
> > > Thanks for carefully considering our response!
> > >
> > > > I think you should include your reasoning for the title from your rebuttal in the paper.
> > >
> > > Will do!
> > >
> > > > Also, I recognize that I underestimated the undercoverage of the CLT in Figure 2, and I would expect many others to do the same. I wonder if you could highlight this difference in the paper in some way, for example by somehow plotting the interval width difference you gave in the rebuttal.
> > >
> > > This is a very good point.  It isn't easy to represent the extent of the undercoverage graphically, while making sure that the plot is still really easy to interpret. But we agree that it is critical to get this right and that interval width is a promising approach.  We will definitely work out how to do this for the camera-ready.

---

### Official Review · Reviewer_o6r6 · 2025-03-10

**Significance:** 2
**Argument Clarity:** 3
**Rating:** 3
**Confidence:** 3

**Questions:**

- Can you explain more about how different priors affect Bayesian credible intervals in LLM evaluations?
- How do the new methods work with bigger data and more complex models? What problems might come up; how can we fix them?
- In your tests, did you find any cases different from your claims where even the suggested methods didn't work well? (If yes, how can these issues be solved?)

**Discussion Potential:**

2

**Paper Summary:**

The paper says we shouldn't use the Central Limit Theorem to make confidence intervals for LLM evaluations. This is because CLT needs a lot of independent samples, which is rare in LLM evaluations. LLM evaluations often have questions that are connected, model outputs that are related, and smaller specialized tests. Also, CLT doesn't work for metrics like F-scores, which aren't just simple averages of independent variables. Its interesting, however, what I really care is the potential soluation of this limitation.

## update after rebuttal

The author's response has addressed some of my concerns, and I will keep the score.

**Position:**

Yes

**Position In Title:**

Yes

**Related Work:**

3

**Strengths And Weaknesses:**

Strengths:
- The paper uses simulations and real examples to show problems with CLT-based confidence intervals in LLM evaluations; It covers an important topic for the ICML community by pointing out issues in current methods and suggesting better options.
- Including practical examples and code snippets makes it easier to use the new methods. this could lead to more discussion and improvements in the field. The paper also refers to related work and discusses different viewpoints.

Weaknesses:

- The paper talks a lot about Bayesian credible intervals but could give more details comparing them with other frequentist methods besides the Wilson score interval and bootstrap.
- While mentioning computational costs, a deeper look at the balance between being computationally efficient and accurate in different methods would make the argument stronger.
-  the paper could expand on how sensitive Bayesian methods are to prior choices and offer more thorough results from various LLM evaluation situations.

**Support:**

3

---

> ### Author Rebuttal · Authors · 2025-03-31
>
> Thanks for your careful and considered review, acknowledging that we cover "an important topic for the ICML community by pointing out issues in current methods and suggesting better options."
>
> ### Comparing Bayesian CIs to other frequentist methods
> In addition to the Wilson score (WS) interval and bootstrap you mention, we also use Clopper-Pearson (Fig 2), and Fisher exact test for the odds ratio (Fig 4). Additionally, in response to reviewer xtBp we’ve added an analysis that uses the Delta method to compute CIs for F1 scores ([Rebuttal Fig 5](https://anonymous.4open.science/api/repo/CLT_rebuttal-CC75/file/rebuttals.pdf?v=c3156a4e)).
>
> Importantly, the point of the paper is not Bayes vs Frequentist. The point is CLT vs non-CLT-based methods, whether Frequentist or Bayesian. We have comprehensively shown that CLT-based CIs perform poorly in the small-data setting, while non-CLT based ones (Frequentist or Bayesian), perform well. Indeed, in Fig 2, we find that the WS interval behaves as well as Bayes which leads to our recommendation: "We would, therefore, recommend using WS or Bayesian intervals in practice."
>
> Of course, if you have a specific additional frequentist method in mind, please let us know and we will add it!
>
> ### Computational Cost
> See [Rebuttal Fig 1](https://anonymous.4open.science/api/repo/CLT_rebuttal-CC75/file/rebuttals.pdf?v=c3156a4e) for an overview of compute time. The key message is that the computational cost of all of these methods is trivial. We ran everything on 1 CPU, and the longest compute time was 200 milliseconds for the largest Bayesian model in Fig 3.  Whilst the cost grows with problem size, these methods are specifically for small data size settings. In the larger data size setting (e.g. > 300) where compute cost might start getting larger, using very fast CLT-based approaches is fine.
>
> ### How do different priors affect Bayesian CIs?
> We use broad, non-informative priors. Specifically, we use a uniform prior over performance (i.e. the probability that an LLM answers a question correctly). That means you can get narrower error bars by putting in extra prior information. For instance, your prior information might be that you expect the model to do pretty well. You might also have a multimodal prior, which might say that you expect the model to either work well or badly, but not get roughly 50% of the questions right (and similarly for skewness and heavy tails). You can build that into an _informative_ (or _subjective_) prior, and hence get narrower error bars. However, using such subjective priors is controversial, with many arguing that the choice of a specific informative prior is ad-hoc and unjustified (see e.g. [1], [2] among others). Indeed, informative priors would be problematic when you're comparing LLMs. You want the error-bars for LLM performance to depend only on the benchmark results, and not on your prior beliefs about how well the models might perform on the data. To reiterate, you really, _really_ don't want your benchmark results to depend on your subjective priors, which may, for instance, be that you expect Llama 3 to do better than Llama 1. As such, using broad, non-informative (also called “objective”) priors seems like the most advisable choice.
>
> Nonetheless, considering the impact of priors is critical in any Bayesian setting, and we do so extensively in Appendix F, where we use our usual _non-informative prior_ for inference, but consider settings where we _could_ build in more prior information.  E.g., we consider settings where the model generally does well (Fig 14) or badly (Fig 13). If you know that, you could build an informative prior, which would give narrower error bars, and "optimal" behaviour in terms of coverage (i.e. actual coverage matches confidence level) and efficiency (i.e. obtaining the narrowest possible intervals). However, you may not know this prior information, or you may choose not to use it for the reasons above. In that case you can't achieve this form of optimality, as getting the correct coverage and maximal efficiency requires knowing the true underlying prior information (see e.g. [3]). Instead, you’d want wider error bars to reflect your lack of potentially useful information, and that's exactly what you get in almost all settings in Appendix F.
> We’ll add a more extensive discussion of these points to the main text.
>
> ### Bigger data, more complex models
> These methods apply equally to LLMs of any size, complexity and trained with any number of tokens. They operate on binary variables representing whether or not the LLM answered a question correctly.
>
> ### Cases where suggested methods didn’t work well
> None that we know of.
>
> ---
> [1] Efron, B. (1986) Why isn't everyone a Bayesian? The American Statistician
>
> [2] Gelman, A. (2008) Objections to Bayesian statistics. Bayesian Analysis
>
> [3] Severini, T. A. (1993) Bayesian interval estimates which are also confidence intervals. Journal of the Royal Statistical Society Series B

---

> > ### Comment · Reviewer_o6r6 · 2025-04-02
> >
> > Thank you for your reply. However, as a position paper, I still suggest discussing some unconventional cases.

---

> > > ### Author Response · Authors · 2025-04-02
> > >
> > > Thanks for carefully considering our response!
> > >
> > > An unconventional case could be an eval that reports accuracy up to $K$ results (i.e. if a model contains the correct output within $K$ generations). This was raised by [reviewer ARZC](https://openreview.net/forum?id=YhZ2PY2nZa&noteId=0Zd2ybH1Ey) and we invite you to look at [our response](https://openreview.net/forum?id=YhZ2PY2nZa&noteId=5aDKzMFa6T) for more details. As mentioned, we have already added a discussion on that point to our working manuscript.
> > >
> > > Please let us know if have any other unconventional cases in mind! We'd be happy to also add a discussions around them.

---

### Official Review · Reviewer_xtBp · 2025-03-13

**Significance:** 3
**Argument Clarity:** 2
**Rating:** 3
**Confidence:** 2

**Questions:**

1. Explore the sensitivity of the proposed methods to different hyperparameter choices, especially for the Bayesian models. A more detailed sensitivity analysis would help readers understand how robust the methods are under various prior specifications.

2. A comprehensive table summarizing the pros, cons, and performance metrics (e.g., coverage, interval width, computational cost) of each method across different scenarios would provide a clear comparative overview for practitioners.

**Discussion Potential:**

2

**Paper Summary:**

This paper critiques the use of CLT-based confidence intervals in large language model (LLM) evaluations, highlighting its limitations due to violations of IID assumptions. It recommends Bayesian credible intervals as a more reliable alternative, particularly in small-sample and non-IID settings. The paper demonstrates through simulations that Bayesian methods outperform CLT and bootstrap approaches, providing better uncertainty quantification. It also includes practical code examples to facilitate adoption. The authors conclude that Bayesian methods are a robust choice for modern LLM evaluation practices.

**Position:**

Yes

**Position In Title:**

Yes

**Related Work:**

2

**Strengths And Weaknesses:**

Strengths:

•	The paper clearly explains CLT principles, its assumptions, and limitations, using mathematical formulas to illustrate failures under Bernoulli and Dirichlet-Categorical models.

•	It recommends Bayesian credible intervals as a reliable alternative for small samples and non-IID data, providing detailed code examples to aid implementation.

•	It summarizes the main issues with CLT methods in LLM evaluations, such as problems with small sample sizes, non-independence, and handling non-linear metrics like the F-score.

Weakness:

•	The experiments mainly use Beta(1,1) and Gamma(1,1) distributions, which may not capture the complex, real-world data characteristics (e.g., skewness, multimodality, heavy tails) seen in LLM evaluations.

•	Although Bayesian methods perform well in small-sample scenarios, their computational complexity might be a challenge for large-scale evaluations. A discussion on scalability and potential efficient approximations would be beneficial.

•	The paper offers minimal discussion on alternative frequentist approaches (e.g., Delta method, Fieller’s Theorem) that might improve upon traditional CLT methods under certain conditions.

**Support:**

2

---

> ### Author Rebuttal · Authors · 2025-03-31
>
> Thanks for your careful and considered review!
>
> ### Computational cost
> See [Rebuttal Fig 1](https://anonymous.4open.science/api/repo/CLT_rebuttal-CC75/file/rebuttals.pdf?v=c3156a4e) for an overview of compute time. The key message is that the computational cost of all of these methods is trivial. We ran everything on 1 CPU, and the longest compute time for any of the methods was 200 milliseconds for the largest Bayesian model in Fig 3. Whilst the cost grows with problem size, these methods are specifically for small data size settings. In the larger data size setting (e.g. > 300) where compute cost might start getting larger, using fast CLT-based approaches is fine.
>
> ### Delta method & Fieller’s Theorem
> Thanks, this is a great idea! We weren’t sure how to compute error bars for complex quantities like the F1 score, as it is a nonlinear function of the average performance, but it turns out you can use the Delta method to compute error bars on the F1-score. We have added these to [Rebuttal Fig 5](https://anonymous.4open.science/api/repo/CLT_rebuttal-CC75/file/rebuttals.pdf?v=c3156a4e), where we find it performs similarly to bootstrap.
>
> It’s important to note that considering approaches such as the Delta method and Fieller’s Theorem do not affect the results in Figures 1-5. That's because the Delta method and Fieller's theorem don't improve CLT-based confidence intervals for standard quantities (e.g. when using $n$ IID samples to estimate the mean $\theta$). And we're working with these standard quantities in Figures 1-5. Instead, they generalise the range of quantities you can compute CLT based error bars for. For instance, the Delta method allows you to give error bars for $g(\theta)$, where $g$ is a nonlinear function (which allows us to get error bars on the F1 score). Fieller's method allows you to take two CLT-based estimates of $\theta_1$ and $\theta_2$ and use them to estimate, and compute error bars on the ratio, $\theta_1 / \theta_2$.
>
> ### Priors, skewness, multimodality, heavy-tails
> We use broad, non-informative priors. Specifically, we use a uniform prior over performance (i.e. the probability that an LLM answers a question correctly). That means you can get narrower error bars by putting in extra prior information. For instance, your prior information might be that you expect the model to do pretty well. You might also have a multimodal prior, which might say that you expect the model to either work well or badly, but not get roughly 50% of the questions right (and similarly for skewness and heavy tails). You can build that into an _informative_ (or _subjective_) prior, and hence get narrower error bars. However, using such subjective priors is controversial, with many arguing that the choice of a specific informative prior is ad-hoc and unjustified (see e.g. [1], [2] among many others). Indeed, informative priors would be problematic when you're comparing LLMs. You want the error-bars for LLM performance to depend only on the benchmark results, and not on your prior beliefs about how well the models might perform on the data. To reiterate, you really, _really_ don't want your benchmark results to depend on your subjective priors, which may, for instance, be that you expect Llama 3 to do better than Llama 1. As such, using broad, non-informative (also called “objective”) priors seems like the most advisable choice.
>
> Nonetheless, considering the impact of priors is critical in any Bayesian setting, and we do so extensively in Appendix F, where we use our usual _non-informative prior_ for inference, but consider settings where we _could_ build in more prior information.  E.g., we consider settings where the model generally does well (e.g. Fig 14) or badly (e.g. Fig 13). If you know that, you could build an informative prior, which would give narrower error bars, and "optimal" behaviour in terms of both coverage (i.e. actual coverage matches confidence level) and efficiency (i.e. obtaining the narrowest possible intervals). However, you may not know this prior information, or you may choose not to use it for the reasons above. In that case you can't achieve this form of optimality, as getting the correct coverage and maximal efficiency requires knowing the true underlying prior information (see e.g. [3]). Instead, you’d want wider error bars to reflect your lack of potentially useful information, and that's exactly what you get  in almost all settings in App. F.
> We'll add a more extensive discussion of these points to the main text.
>
> ### Summary Table
> Agreed. We’ve added [Rebuttal Table 1](https://anonymous.4open.science/api/repo/CLT_rebuttal-CC75/file/rebuttals.pdf?v=c3156a4e).
>
> ---
>
> [1] Efron, B. (1986) Why isn't everyone a Bayesian? The American Statistician
>
> [2] Gelman, A. (2008) Objections to Bayesian statistics. Bayesian Analysis
>
> [3] Severini, T. A. (1993) Bayesian interval estimates which are also confidence intervals. Journal of the Royal Statistical Society Series B

---

### Official Review · Reviewer_ARZC · 2025-03-16

**Significance:** 4
**Argument Clarity:** 4
**Rating:** 5
**Confidence:** 4

**Questions:**

In practice, I rarely see people report confidence intervals (or credible intervals) and compare their performance only with the sample average of metrics like accuracy. This paper shows a good direction for improving results reporting and offers a simple formula and code. Could you provide a concrete application of the proposed position in LLM evaluation benchmarks?

What would be the limitation of applying this kind of statistical inference on LLM evaluations?

There is a popular metric that reports accuracy up to K results. Namely, if a model contains a correct output with K generations. In this scenario, what would be a good way to apply the proposed approach?

**Discussion Potential:**

3

**Paper Summary:**

The position of this paper can be stated as follows.
In language model evaluations, let's use Bayesian credible interval or other frequentist methods like Wilson instead of CLT.
The main motivation is that the number of independent samples can be small in practice, and real-world benchmarks often don't satisfy assumptions for CLT.

Section 3 shows several scenarios with an experiment setting with 200 independent datasets of different sizes.
The simulation study shows that Bayes approach matches well to the 1-$\alpha$ coverage when N is smaller.

**Position:**

Yes

**Position In Title:**

Yes

**Related Work:**

3

**Strengths And Weaknesses:**

The strength of this paper is on identifying 5 notable scenarios that would commonly appear in LM evaluations.
The numerical experiments support the authors' position and claims well.

The weakness of this paper would be that the experiment is a simulation study that doesn't reflect actual LM benchmark evaluations.

**Support:**

4

---

> ### Author Rebuttal · Authors · 2025-03-31
>
> Thanks for your careful and extremely positive review, acknowledging that "The strength of this paper is on identifying 5 notable scenarios that would commonly appear in LM evaluations. The numerical experiments support the authors' position and claims well."
>
> > Could you provide a concrete application of the proposed position in LLM evaluation benchmarks?
>
> Please see Figure 1, which gives error-bars on the LangChain tool-use benchmark. Note that this was inspired by a “real-world” failure of the CLT [in this Langchain blog post](https://blog.langchain.dev/benchmarking-agent-tool-use/).
>
>  We have also added another plot showing error-bars on a “real-world” evals dataset: [MathArena AIME 2025 II](https://matharena.ai/) – see [Rebuttal Fig 6](https://anonymous.4open.science/api/repo/CLT_rebuttal-CC75/file/rebuttals.pdf?v=c3156a4e). Here with $N=15$ we see CLT-based intervals collapsing to zero-width and extending past $[0,1]$.
>
> > There is a popular metric that reports accuracy up to K results. Namely, if a model contains a correct output with K generations. In this scenario, what would be a good way to apply the proposed approach?
>
> This is a really interesting question.
>
> We certainly could get error bars for this approach.  The basic idea would be to take a single model and a single question, generate N times, and find the number of generations that give correct answers.  Then you can infer a Bernoulli posterior (with error bars) over the probability of a single generation being correct.  You can use this to compute the error bars for the probability of at least 1 in K generations being correct.
>
> The above gives principled error bars for the performance of a single model on a single question. But we're really interested in a single number that measures the performance of a model on all the questions in a dataset.  To do that, you need a hierarchical Bayesian model with a per-model latent variable, which captures the average probability of a single generation being correct on a single, randomly chosen question. This isn't trivial, but it is reasonably straightforward, and we have all the tools required for efficient inference.
>
> As such, we will add a discussion of this to the camera-ready, but it isn't clear to us whether we can compare against CLT based methods in this setting, as we aren't actually sure what CLT-based method would be appropriate.
>
> Finally, addressing the weakness that you note:
>
> > The weakness of this paper would be that the experiment is a simulation study that doesn't reflect actual LM benchmark evaluations.
>
> The main reason we conduct an extensive simulation study is to be able to demonstrate the properties of different methods (namely coverage, interval length), which cannot be evaluated if we don’t know the ground truth. However, as mentioned earlier, Figure 1 gives an example of how some of the issues we discuss do appear in practice. We’ll additionally include the MathArena figure mentioned above when revising the paper.

---

> > ### Comment · Reviewer_ARZC · 2025-04-02
> >
> > Thanks very much for your response.
> > The code in the supplementary material will be a very useful tool for empirical researchers in this area. Most of the time, people don't consider error analysis or use average or simple CLT-based error bars, as shown in Figure 1.
> > This practice could also be applicable to other empirical research areas like reinforcement learning, where people usually report the average of the performance curves over 5 to 20 trials, which often introduces debates about whether it is significant or not.
> > I will recommend this position paper to my colleagues and apply the methods suggested in this paper rather than reporting raw numbers.
> >
> > I also read responses from other reviewers, and this paper provides a good starting point for discussing what statistical methods would be more appropriate for reporting LLM experiments.
> > The topic is well aligned with the position track ("How we can improve the ways that we conduct and evaluate machine learning research") and it has a good discussion potential ("The topic is likely to inspire constructive, useful discussion within the ICML community. The reviewer need not agree with the stated position.").

---

> > > ### Author Response · Authors · 2025-04-02
> > >
> > > Thank you very much for recognising the value of our work and highlighting its relevance empirical research more broadly.
> > >
> > > We are glad to hear that you found the supplementary code useful. We will be releasing a lightweight library with the camera ready version, which we hope will make it even easier for researchers to add error bars to their analysis.

---

### Decision · Program_Chairs · 2025-04-30

**Decision:**

Accept (spotlight poster)

**Comment:**

The position of the paper is clearly relevant for frontier LLM evaluation. The proposed alternatives  to CLT-based CIs are clearly better and easy to implement. All reviewers are supportive of this paper. I would recommend a strong-ish accept.